

# Key learning moments as predictors for understanding snowpack dynamics during a season-long avalanche course?

Tim Dassler[1], Richard Fjellaksel[1,2], Gerit Pfuhl[1,3]

[1]Department of Psychology, CARE Center for Avalanche Research and Education, UiT The Arctic University of Norway, Tromsø, 9037, Norway
[2]Department of Health and Care Sciences, UiT The Arctic University of Norway, Tromsø, 9037, Norway
[3] Department of Psychology, NTNU Norwegian University of Science and Technology, Trondheim, 7491, Norway

*Correspondence to*: Tim Dassler (tim.dassler@uit.no)

**Abstract.** When skiing in avalanche-prone backcountry terrain, the thrill of the experience is always accompanied by the looming risks of injury or worse. Safety in snowy mountains hinges on understanding avalanche dangers. Avalanche education serves as a crucial tool to impart the skills needed to navigate these hazards safely. Understanding snowpack dynamics, i.e.,

whether the snow is stable, and conditions are safe enough is pivotal for making high quality safety judgments in potentially hazardous avalanche terrain. In our innovative and explorative avalanche course study, participants' learning was monitored throughout an entire ski season and assessed a year after the course. Participants were co-designers, with their experiences and reflections forming course content. We measured learning outcomes before, during and a year after by collecting both qualitative and quantitative data from course participants. We present quantitative data from surveys assessing Key Learning

Moments, challenges and skills, as well as from a log assessing knowledge about snow factors ("AviLog"). Our results show that the number of learning moments participants reported can indicate learning outcomes. However, we find that learning does not last for all. We discuss implications for mountain safety and avalanche education.

KEYWORDS: Decision-making, avalanche education, snowpack assessment, avalanche prevention, avalanche risk,

experiential learning






## 1 Introduction

"Data is not information, information is not knowledge, knowledge is not understanding, understanding is not wisdom."

- Clifford Stoll

Winter backcountry recreation brings inherent risks, particularly in mountainous regions where snow avalanches pose a significant threat to backcountry recreationists. Avalanches are among the most serious natural hazards in mountain environments. They can be triggered by a single individual and claim an average of 38 fatalities in North America (Avalanche-center.org, 2024) and 100 lives annually in the European Alps (European Avalanche Warning Services, 2014). In most cases

people expose themselves voluntarily to the hazard (Techel, 2015). Official records from Norway indicate that 826 individuals have been caught in avalanches since 2008 (Varsom.no), with 108 fatalities during the same period, though underreporting likely leads to an underestimation of these figures.

The vast majority of avalanche accidents are triggered by victims themselves or their companions (Schweizer and Lütschg, 2001; Rainer et al., 2008), underscoring the potential for prevention through improved avalanche education. Avalanche courses

aim to equip individuals with essential skills to assess danger and travel safely in potentially hazardous terrain. The introductory quote highlights the challenge of avalanche education to integrate theoretical knowledge with practical skills, which are crucial for making sound judgments in snowy and potentially hazardous terrain. It emphasizes the hierarchical relationship between different levels of cognitive processing, progressing from simple data to wisdom—the ability to make informed decisions based on understanding and experience.

Despite the increasing popularity of winter recreation in avalanche terrain and growth in certified avalanche courses, research in this field is sparse. Only a handful of peer-reviewed studies have focused on avalanche education, often exploring specific topics or drawing implications from research in other educational domains (Fisher et al., 2022; Greene et al., 2022; Johnson et al., 2020; Zajchowski et al., 2016;) but not investigating what the participant actually learns from a course and is able to apply in the field. There is a clear need for research in avalanche education to address its distinct challenges and inform effective

teaching and learning practices. Furthermore, there is a substantial lack of knowledge regarding the effectiveness of avalanche training.

Indeed, avalanche hazard cannot be taught by description alone (as for radiation or chemical toxins) given that on the one hand people seek out nature and the beauty, and on the other hand the hazard is nearly impossible to estimate and predict with certainty (Landrø et al., 2022). Crucially, avalanche courses are a popular means to learn about avalanche risk but what is

taught is rarely perceived as relevant by experts (Landrø et al., 2020).

### 1.1 What is learning?

Alexander et al. (2009) describe learning as "a multidimensional process that results in a relatively enduring change in a person or persons, and consequently how that person or persons will perceive the world and reciprocally respond to its affordances physically, psychologically, and socially." (p. 186). Thus, learning is fundamentally rooted in a systemic, dynamic, and





interactive connection between the learner, the type of learning and its subject matter. This connection is ecologically placed in a specific context and period, and evolves over time. Change in how a person "perceives the world" is a pre-condition to reacting to this changed perception (Alexander et al., 2009) and changing one´s behavior. There are three important features from this definition we explore in our study of course participants' learning of snowpack dynamics. (1) Do we see a change in understanding of snow factors? (2) Can participants use this changed understanding in their own practices, that is, when judging

the safety of a slope, and (3) is this a lasting or stable change that persists after the course?

Given that the assessment of avalanche hazard is a complex topic, and that learning is happening in a wicked learning environment (Hogarth et al., 2015), shallow learning, using rules of thumb, or relying on statistics will not suffice. To allow flexibility in adapting to changing snow conditions, a "deeper" understanding of snowpack dynamics is required.

## 1.2 Experiential and authentic learning in complex and wicked environments

The importance of applied and experiential learning is well established in professional fields, where the integration of theory and practice is particularly emphasized, along with achieving a deep understanding of the concepts, models, and methods relevant to the field (Dettlaff & Wallace, 2003). This includes both internal competence—referring to a thorough understanding of the internal dynamics of the field—and practical competence, which involves applying these concepts, models, and methods to real-world problems. A key aspect here is the ability to tackle partially unknown, incomplete, and complex challenges in

new situations, rather than merely solving standardized and predefined problems. Experiential learning is also an integral part of outdoor education and adventure learning due to its key advantage closer to the events and actions, which is argued to result in a more impactful learning experience (Beames, 2016).

This is also reflected in the curriculum for avalanche courses in Norway as defined by the Norwegian Mountain Forum (NF) that puts a prime on experiential and authentic situational learning and specifies that "*The course focuses on the touring*

*situation […] As much of the course as possible should be conducted outdoors, with theoretical elements taught in connection to situations where they naturally fit.*" (own translation) (Norsk Fjellsportforum, 2018a).

Thus, learning theories that emphasize experiential learning, how experience is transformed into a change in knowing, thinking, feeling and doing through the interaction between the learner and the environment, are especially relevant when discussing avalanche education. Experiential learning theory, as introduced by David A. Kolb (1984), is a prominent learning theory that

emphasizes the role of experience in knowledge creation and transformation. It integrates several elements from the works of John Dewey, Jean Piaget, and Kurt Lewin and their foundational theories on learning in psychology and education. According to Kolb, learning is a cyclical process that involves four stages: concrete experience, reflective observation, abstract conceptualization, and active experimentation (Murrell and Claxton, 1987). This experiential learning cycle, a key concept in Kolb´s theory, has been widely used in educational programs to actively engage learners (Kolb and Kolb, 2018). It is

particularly beneficial for adult learners, as it encourages self-directed learning and real-world problem-solving (Dernova, 2015).





Transformative learning theory, introduced by Jack Mezirow (1978a, 1978b), is another relevant learning theory focusing on how experiences are transformed into new ways of knowing, thinking, feeling and acting (Calleja, 2014). Learning for Mezirow starts with a disruptive experience or moments of dislocation where the learner experiences a form of dissonance between their beliefs and the world, for example, a discrepancy between what they think they do and what they actually do. An archetypal example of such a disruptive experience is unintentionally being caught by an avalanche, which is usually pretty good feedback on that you were doing something else than you thought you were doing. Such dislocatory moments can then become triggers for reflection and transforming the learners understanding and perception of the world; including a revision of our mental maps or schemas (Argyris and Schön, 1974) and a change of how we think and feel about the world. Both experiential and transformative learning theory include experience and the reflection upon the experience that leads us to form new understandings of the world which affect how we act in the world as essential parts of the learning process.

To promote effective learning, Biggs (2011) emphasizes the importance of learning from *relevant* (own emphasis) experiences and thus structuring teaching to closely align with "authentic situations". Authentic situations are contexts that learners will encounter in everyday life and practice. Experiential learning building on relevant experience in authentic situations provides real and contextual experiences and is particularly relevant for avalanche education. Yet, achieving the intended learning outcomes within the set timeframe of a single avalanche course (or two) can be challenging (Landrø et al., 2022). Some of the drawbacks are that experiential learning that is tied to first-hand experience of learning situations, also called situational learning (Hallandvik and Høyem, 2019), can often be unpredictable in a dynamic environment such as snowy mountains. The repetition necessary for effective learning is both time-consuming and sometimes impractical due to the ever-changing and sometimes hazardous conditions. The necessary conditions to practice or demonstrate a specific situation or skill may be lacking or too dangerous. Furthermore, the artificiality of the learner group composition, choice of learning terrain and snow conditions may also make it difficult to create truly authentic learning situations.

## 1.3 What are Key Learning Moments and how to capture them?

How can one facilitate learning such a complex matter as the interplay of various snow factors on snowpack stability? As a first step, we set out to identify moments when learning happens for the learner. The aim is to know what the learners experienced as significant and intensive learning moments, and what was going on in these situations that they could remember and report. Such events significant for learning are known as Key Memorable Events or Dislocatory Moments.

Key memorable events connect to experiential learning theory and are learning moments that are experienced as 'a-ha' moments where new pieces of the puzzle fall into place, or a new understanding takes hold in the learner. These moments are experienced as intense and often meaningful moments, with strong feelings attached to them, and can have a significant impact on students´ learning (Marmur, 2019).

Dislocatory moments can be considered a special kind of key memorable events. These are critical events that disrupt an individual´s usual understanding of the world, potentially leading to significant changes in thinking, feeling or behavior. They are thus connected to transformative learning theory. These disruptions challenge deeply held beliefs and prompt critical



reflection, often arising from new experiences, such as exposure to new ideas or where people experience a discrepancy between what they think they do, and what they actually do (Hesjedal, 2020; Schön, 2017). The discomfort experienced during dislocatory moments can act as a catalyst for transformative change (Davison, 2020; Gilbert, 1997; Laros, 2017) in the form of changing how one thinks, feels about, and acts in the world.

Since we were interested in capturing both types of learning moments, we asked participants to send in reports of their own

experienced intensive learning moments (for details see Fjellaksel et al., 2024, in preparation) as well as what they felt, thought and did, and if that had any notable consequence. We call these Key Learning Moments. Key learning moments that were reported by the participants include the following four criteria: (1) being intense, (2) being revelatory or dislocatory in some way, with (3) some kind of thought, feeling or action associated with them, and (4) indicative of a change in understanding, thinking, feeling and/or doing. Our definition of key learning moments includes both key memorable events as well as

dislocatory moments.

**1.4 A season-long intensive avalanche course**

The aim of this study is to investigate the impact of key learning moments on participants' understanding of the relationship between snow factors and snowpack dynamics, including the stability or instability of the snowpack. Additionally, the study explores whether participants can effectively identify, assess, and incorporate these snow factors into their sense- and decision-

making processes. Furthermore, the research aims to evaluate the longevity of any changes in participants' abilities and understanding over time.

Since doing and learning to do snow stability and avalanche danger assessments in complex situations and wicked environments requires detailed observations, being able to assess and judge these observations and carefully weighting the various relevant risk factors and uncertainty (Landrø, et al., 2022; McClung and Schaerer, 2006; Reuter et al., 2020) it is akin

to a form of abductive reasoning; the process of inferring the most plausible explanation for a given set of observations (e.g. Walton, 2004). It is this real-world application that is definitive for what we look for in the data. We are not only interested in if participants have gained new theoretical knowledge about snow factors, that is, do they know the snow factors. Rather, we also look for whether participants understand the factors´ relevance for assessing snowpack stability and are able to apply this knowledge in their judgment whether a slope is safe to ski under the current conditions. Finally, we look at if participants can

retrieve their reasoning post trip and are able to use it in justifying their choices.

Given the lack of studies in the effectiveness of avalanche courses and their long-term effects, we designed a season-long course with participants being co-designers. The aim of the study was manifold. Firstly, by participants being co-designers we aimed to teach not only what we thought they should know, but what the participants felt most need for. Secondly, by using both qualitative and quantitative methods we aimed to identify what is learned during a season-long course. Thirdly, we wanted

to look at the effects of taking an avalanche course over time. Both during the course, and a year after. Through a follow up study where we followed the participants on ski tours in avalanche terrain a year after the course, we wanted to look at which of the learning that took place at the course lasted and was implemented in their own ski touring practices.



While our overarching interest was what was learned during the course and if there were lasting effects, in that the acquired knowledge was implemented in the participant practices or that it became a habit for them, we here focus on quantitative data (for an analysis of the qualitative data please see Dassler et al., 2023).

To explore these questions, we asked participants about their challenges, competencies and what they want to become better at, before and after each module as well as their confidence to rely on their judgements of snow stability. They reported their key learning moments and filled out a detailed log to assess their understanding of snow factors (modified from Landrø et al., 2020). Importantly, in the skill survey we asked them to justify why it was safe to ski the slope they skied during the trip.

We formulated the following key hypotheses:

H1: There is an improvement in knowledge across the course and the newly acquired knowledge is also evident after one year.

H2: The more key learning moments within the domain of snow and terrain, the more there is an improvement in knowledge of snow stability.

H3: There is no improvement of knowledge but a larger awareness of one's ignorance or gaps in knowledge (increase in epistemic uncertainty).

If H1 is true it follows that avalanche education (as regulated by the Norwegian Mountain Forum) leads to a long-lasting understanding of avalanche danger and risk. If H2 is true it follows that a priori understanding of hazards can accelerate mastering the complexity of avalanche danger and risk. If H3 is true, the seeds for self-learning and experiential learning are sown.

## 2 Methods

Since we were interested in whether participants could learn how to use the assessment of snow factors in their judgment of snowpack stability and the justification of their subsequent decisions, we chose a knowledge-based analytic approach toward teaching avalanche conditions assessment that was inspired by the Systematic Snow-Cover Analysis (Kronthaler and Mitterer, 2014; Kronthaler, 2019) where snow factors and layers in the snowpack are identified, analyzed and judged for their likelihood of leading to avalanches. This approach was continually contrasted, supplemented and taught along a more test-based approach that focused on stressing the snowpack and testing for instabilities through both formal stability tests such as the Compression Test (CT), Extended Column Test (ECT), Propagation Saw Test (PST) and Rutschblock Test, as well as informal tests such as the "Burp Test", Handshear tests, and tests where ski poles were used to probe for weak layers.

To assess the effectiveness, participants filled out two types of surveys (see section 2.2.).

### 2.1 Procedure

#### 2.1.1 Participant selection and recruitment process

For the study, participants were selected through the CARE panel (2023), part of a longitudinal research project analyzing behaviors in backcountry environments. The recruitment process involved practical considerations to ensure that participants



were physically capable of engaging in the course activities. Recruitment was geographically confined to the Troms and
Finnmark region in Northern Norway. Potential participants were provided detailed information about the course content and
the physical attendance required before agreeing to participate. Out of the total, 29 individuals consented to participate, with
22 joining through the CARE panel and 7 contacting us directly. We targeted individuals who self-identified as having low to
moderate proficiency in evaluating avalanche risks and possessing average skiing abilities suitable for navigating through
avalanche-prone areas, including both release and trigger zones. We extended an offer to 16 candidates to join the course, 12
accepted. However, 2 withdrew before the course began due to logistical and medical reasons, resulting in a final group of 10
participants (8 men and 2 women).

**2.1.2 Course structure**

The course was structured around a modular framework, comprising seven practical days divided into four modules. It was
part of a season long research project on learning at avalanche courses. Starting in January, with additional modules in February
and March the initial session focused on introductions and a preliminary outing (baseline trip) to observe the group's behavioral
dynamics in avalanche-prone areas. Each subsequent module encompassed two practical days, with one day allocated to indoor
learning due to adverse weather conditions. This setup facilitated a deep dive into the theoretical and practical aspects of
snowpack analysis, interpersonal dynamics, companion rescue, and strategic planning for navigating avalanche terrain. The
course's first four modules adhered to the curriculum standards set by the Norwegian Mountain Forum (Norsk Fjellsportforum,
2018b) for basic (Level 1) and advanced (Level 2) avalanche training. We also invited participants back for two consecutive
modules (each 2 days) where they had to apply all they had learned during the first four modules by planning and executing
ski touring trips as a group. These days took place in April and May 2023. A year later we invited the participants back for a
follow up module (two separate days, March 2024) to check how much they still remembered from the previous year and were
able to implement in their own practices. The choice of mountain and learning environment was made by the instructors during
module 1 and 2. While participants progressed, trip choices were made through shared decisions during module 3 and 4 and
decided solely by the participants in all consecutive modules, both toward the end of the season-long course and the year after.
A typical course day for participants spanned from 8:30 am to 4 pm, encompassing travel to the mountain, pre-trip briefings,
ski tours emphasizing various aspects of avalanche safety, and post-trip debriefings. For the instructors and researchers, days
commenced with a safety briefing, followed by participation in the day's activities and post-debrief notetaking to capture
observations and insights.
Participants engaged in reflective exercises and questionnaires after each module and completed a skills survey (AviLog) after
the first day (baseline trip), the last day of module four of the course (i.e., after finishing the curriculum of the NF advanced
level 2), and after planning and completing a tour in avalanche terrain a year after the course, aiding in the evaluation of
learning outcomes and skill development. Participants were encouraged to fill out surveys and questionnaires within 24 hours
after a course module.





### 2.1.3 Ethical considerations, health and safety

A comprehensive Health, Safety, and Environment (HSE) strategy was developed and implemented, including detailed risk assessments for each module and training on risk mitigation and emergency response. Legal considerations were addressed to clarify liability and financial responsibilities in the event of an accident. The study received approval from the Norwegian

Centre for Research Data (SIKT) and ethical clearance from the institutional review board at the Department of Psychology at UiT The Arctic University of Norway, ensuring compliance with legal and ethical standards.

### 2.2 Material

There are five data sets capturing various aspects of learning during the course. (1) Participant survey responses to the modules, (2) skill surveys (AviLog), (3) Participant observations of their own key memorable learning moments, (4) Embedded observer

and participant observer (instructors) field notes and reports, and (5) Participant focus groups. In this paper we focus on the analysis of participant survey responses (1) and (2), as well as KLMs extracted from (3). The qualitative analysis of findings for data set (4) and (5) is presented in Dassler et al. (2023)

### 2.2.1 Module surveys and participants key memorable learning moments (data sets 1 and 3)

There were two kinds of module surveys, one after each module and one prior to the follow up modules. After each module

participants were asked "To what extent do you feel you have the skills to be able to travel safely on terrain steeper than 30 degrees on stratified winter snow?" and "How much do you trust your own judgments when assessing the stability of the snow cover?". Both were rated on a 7-point Likert scale from 1 = "not at all" to 7 = "very much". Furthermore, participants were asked about experienced challenges ("What did you think was challenging about the two days on the course?"), acquired competencies ("What competencies do you think you have now gained enough proficiency in?") and required support ("What

do you need right now or what can help you move forward?"). These text answers were analyzed and classified by domain, i.e. was the challenge, competency or support in the domain of snow, terrain, weather, gear, human factor or other. After each tour participants reported their key memorable events on a sheet of paper (Fjellaksel et al., 2024, in preparation). They drew a curve indicating the intensity of their experienced learning moments, including what happened, what they thought, felt, did, and if the learning moment lead to any consequences, e.g., changing their behavior or making plans to change their behavior.

The answers were classified by domain. For the follow up study, we used a participant survey asking the same questions as above prior to the follow up module to map if self-rated competence and confidence as well as challenges had changed. Participants filled out the same post module surveys after the follow up module.

### 2.2.2 Skill survey AviLog (data set 2)

The AviLog is a logging survey that assesses demographics, factors relevant for a backcountry trip, including the level of detail

of planning sources, e.g., avalanche forecast, mindset and goal for the trip, group dynamics and 23 snow factors related to a



systematic approach of snow cover analysis. Demographic items were a) gender, b) year born (age), c) years of backcountry skiing, d) days of backcountry skiing per season, e) avalanche education (from none to professional avalanche training). We asked about the source and level of planning employed; a) avalanche forecast, b) weather forecast, c) observational database, d) talking to other people, e) own tracking of the snow and weather history, and for each item answer options were: no detail,

low detail or high detail. For 23 snow factors (see Landrø et al., 2020) we asked whether the factor was assessed and if not why not, the frequency of updating (not analyzed) and if the observation of the factor affected the assessment of snow stability. If the factor was not assessed answer options were: "I don't know what this factor is", "I know the factor but cannot assess it", "I know how to assess it, but don't know how it affects snow stability", and "The factor was not relevant during our trip". We did not analyze detailed observations from the snow stability test. For the 23 snow factors we calculated how much the

participants' choice of factors agrees with that of the instructor(s), how often they answered not knowing the factor (don't know, know but cannot assess it, don't know how it affects snow stability).

We measured understanding of why it is safe to ski with the item: "Why did you decide to ascend / descend the most critical slope? (please write a very short summary of your reflections)," or "Why did you decide not to ascend / descend the most critical slope?". The "why to ski" answers were scored by two independent raters (T.D., R.F.) on whether the reason provided

aligns with the systematic snow cover analysis (scored as 1, e.g., "I considered the slope to be safe enough. The SSA [systematic snow cover analysis] revealed weak layers in the snow cover, but no propagation in these. In addition, it was very difficult to affect the weak layers far down in the snow cover."), partly aligns with it (scored as 0.5, e.g., "It was considered safe. Tests (SBT and ECT) conducted at the top of the slope were crucial for the decision to descend.") or did not align with it (scored as 0, e.g., "The snow seemed promising, and we found no major challenges with it."). In case of discrepancy in the

scoring the two raters discussed their scoring, and the agreed upon scoring was used.

**2.3 Analysis**

For hypothesis 1 we investigated a) whether there is any change in how detailed they planned a trip, b) knowledge about the snow factors, c) applying it to justify descending the most critical slope ("why to ski"), whereby all data came from the AviLog, and c) changes in self-rated competency, measured in the module surveys. We used linear mixed models or Chi-square tests.

Regarding hypothesis 2 the "why to ski" score was the outcome and the amount of reported key learning moments as well as the module (baseline, after four modules or after one year) were the predictors. We also explored whether there was any relationship between the "why to ski" score, accuracy of identifying the relevant snow factors, number of not knowing what the factor measured, reported challenges and competencies.

For hypothesis 3, we related the self-rated confidence scores to their snow stability scores and the number of reported

challenges within the snow and terrain domain with the snow stability scores. We used Kendall's tau as correlation coefficient. Analysis was done in the statistical environment R (R Studio, 2023).



## 3 Results

10 participants (2 women), age ranging from 25 to 68 years, with 1 to 15 years of experience with backcountry skiing (three with one year, two with four years, and one each with two, three, seven, nine or 15 years), and spending many days per season 290 skiing in avalanche terrain (two up to 30 days, six 31 to 60 days, and two 61 to 90 days per season) took part. Among the participants, three reported no formal avalanche education, six reported previous attendance of an avalanche course level 1, one reported completion of avalanche course level 2, and two reported some other avalanche education. All participants took part in the season-long course, and eight took part in the follow-up module one year later.

The participants being co-designers of the course meant that they to a degree were responsible for and could influence the 295 educational content and approach we chose for the course. 8 out of 10 participants explicitly expressed the desire to systematically learn more about snowpack dynamics in order to being able to "*find and ski good snow in steep terrain more often*" in a way that was justifiable, in other words, in a way that was safe enough. One of the remaining two participants explicitly stated that he was not motivated to seek out steep terrain, but that he was highly motivated to learn more about snowpack dynamics in a structured way. This is partly reflected in the participants' answers to what the main goal of their trip 300 was. After the baseline trip all participants reported (multiple choices possible) that their main goals were the course and a social focus, two others reported a skiing focus, four enjoying a nice view and two experiencing nature. After module 4 eight reported skiing focus, two reported snow and avalanche observation focus and four reported a focus on a specific run. Similarly, one year later seven participants reported skiing focus, six a social focus, and three a snow and avalanche focus. Two reported also a specific peak, nice view and one experiencing nature. Thus, snow and avalanche observation as goal was only reported 305 by two and three participants after module 4 and one year later, respectively.

Hypothesis 1 investigated whether there is a) a change in planning sources and details (Fig 1), b) improvement in snow factor knowledge and understanding from the baseline tour to the next season (Fig 2).

At baseline 7 out of 10 participants used high details from the avalanche forecast, 2 used low details and one reported using no details. Regarding the other sources participants did rarely acquire highly detailed information by talking to other people, 310 tracking the snow and weather history, using the observation database (Varsom.no) or the weather forecast. This changed during the course but not all participants continued with it. Across all five sources there was a significant change from baseline to module 4 and one year later in the proportion of using high detailed information; $X^2(df = 4) = 10.132$, p = .038, Cramer's V = .194. For each source the chi-square test was not significant, but the effect sizes were between .21 and .36, see inset in figure 1, particularly participants make more use of tracking the snow and weather history and the observational database 315 (Varsom.no). Overall, we found that they acquire more details about avalanche conditions from more sources.





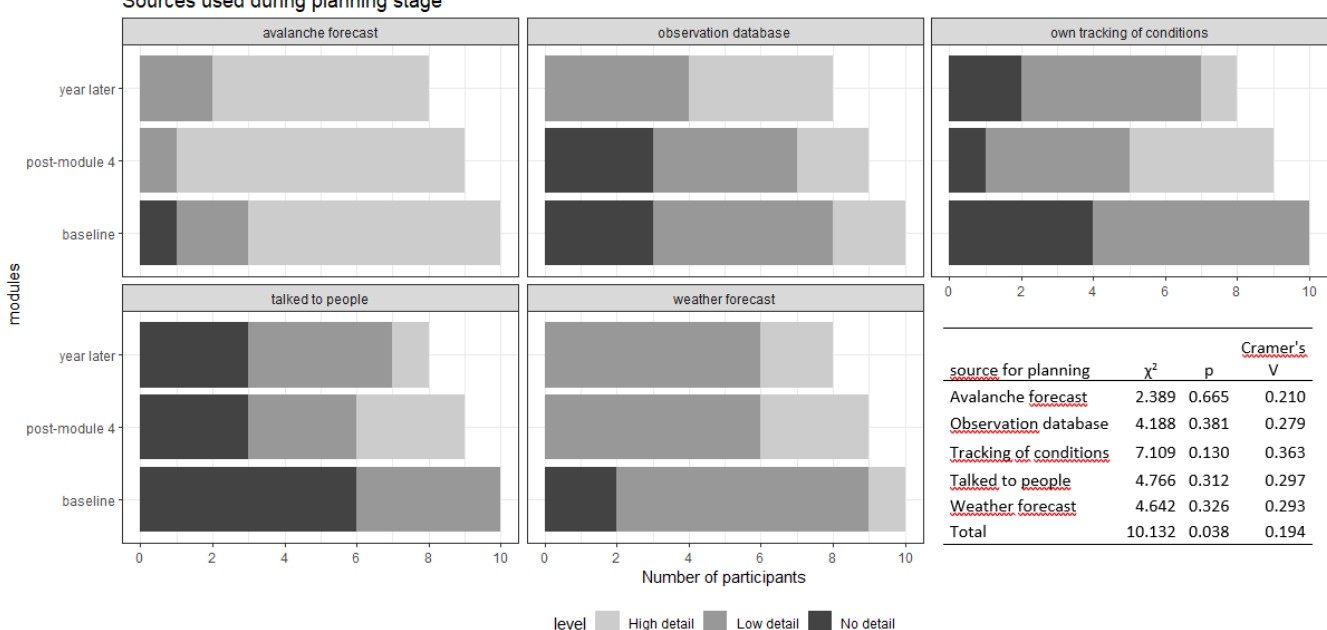

**Figure 1.** Sources used during planning stage. Across five sources participants indicated how much detail they acquired. For avalanche forecast high detail refers to reading the whole forecast whereas low detail refers to checking the danger level and type of avalanche problem. For weather forecast high level refers to reading the detailed forecast, checking radar or windy.com whereas low detail refers to checking the symbolized weather forecast on e.g., yr.no. For snow observation database high level refers to reading several observations for the area in detail whereas low level refers to scrolling through it and looking at pictures. For talking to people with knowledge about the local avalanche conditions, high detail refers to e.g., talking about distribution and reactiveness of the weak layer whereas low detail refers to e.g., talking about "good" snow and snow stability. For own tracking of snow and weather history high detail refers to own snow and weather analysis whereas low detail refers to e.g., absence and presence of previous avalanche activities. Abbreviations: ava. ... avalanche, cond ... condition, db ... database

Participants indicated better understanding what the various snow factors are and how to assess them (Fig 2A), i.e. there were fewer "don't knows" and "know the factor but not how to assess it" after module 4 and one year later (from $M_{baseline}$ = 3.6 to $M_{post4}$ = .8 to $M_{1y}$ = .62, given the small sample size this difference failed to reach statistical significance, all p's > .1). There was also a statistically significant improvement in providing justified explanations for why it was safe to descend the critical slope (Fig 2B) over the duration of the course ($M_{baseline}$ = .25, $M_{post4}$ = .85, p < .001), however, this improvement did not last into the next season ($M_{1y}$ = .25, p = 1). There was a non-significant improvement in selecting the relevant snow factors (Fig 2C) from baseline ($M_{baseline}$ = -.19) to module 4 ($M_{post}$ = .33, p > .5), but this was not maintained one year later ($M_{1y}$ = .22). Regarding self-rated competencies, participants rated their competency on a scale from 1 to 7 at 3.9 at baseline, after four modules at 4.5 and one year later at 4.9 (Fig 2D). This change was statistically significant (baseline to after module 4: p = .01853, baseline to 1y later: p = .00224). Overall, we find that participants do improve in their knowledge of snow factors, although not all knowledge is applied after one year.



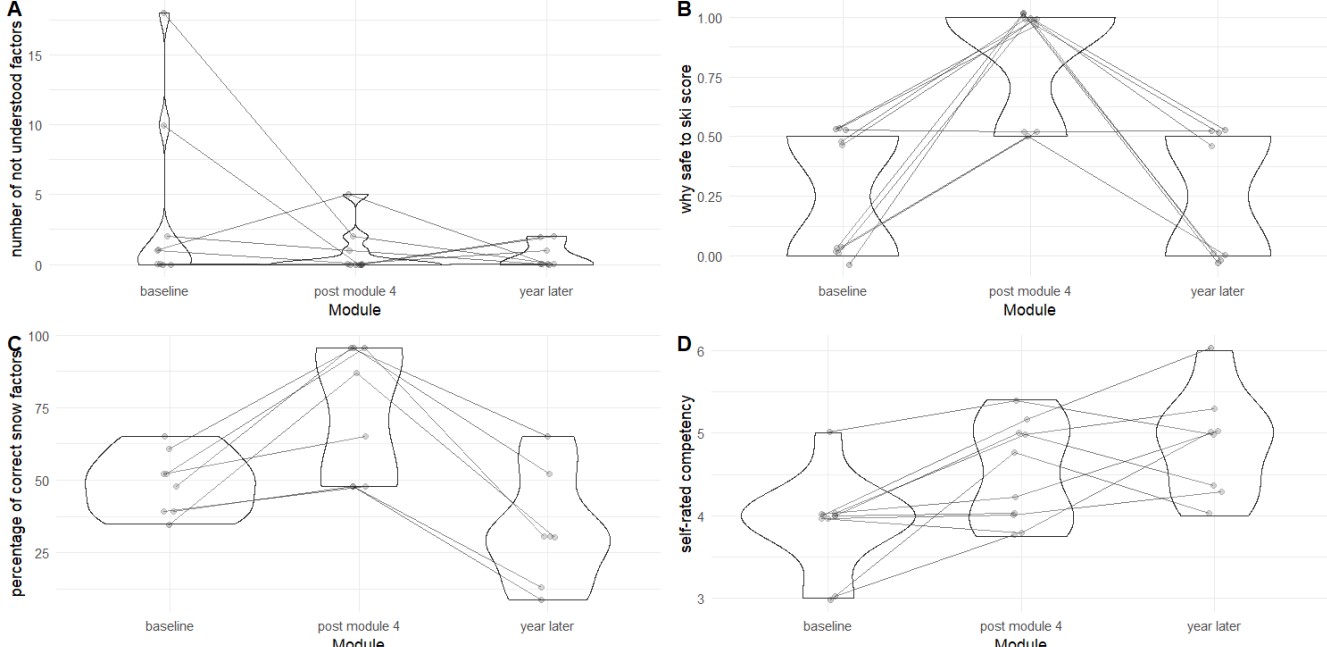

**Figure 2.** Learning of snow factors assessed during the study. Panel A shows that lack of knowledge about snow factors decreases during the course and this improvement remains one year later. Panel B shows a violin plot and the increase and then decrease in reasons given for why it is safe to descend the critical slope. Panel C shows the percentage of accuracy, i.e., selecting among the 23 snow factors the relevant ones. Panel D shows self-rated competency on a 7-point Likert scale. Note that for three participants self-rated competency decreases at the one-year follow-up, whereas five participants reported a slight increase in competency. Lines represent the 8 to 10 participants and are jittered for visibility.

We next investigated whether the "why to ski" score was related to the number of reported key learning moments within the domain snow and terrain (hypothesis 2) and or whether it was related to self-rated confidence or challenges (hypothesis 3). We found that the more key learning moments per module were reported, the less the "why to ski" score improved ($\beta$ = -.187, p = .056), i.e., participants who experienced many key learning moments were not yet mastering a justified explanation of why it is safe to descend the critical slope after module 4 (Fig 3). This may indicate some implicit knowledge of one's own lack of understanding.

We next investigated whether the "why to ski" score after module 4 and 1 year later was related to self-rated confidence or the number of reported challenges for snow and terrain. There was a trend towards higher confidence being associated with a higher (better) "why to ski" score ($\beta$ = .172, p = .083). There was no association between reported challenges and the "why to ski score" ($\beta$ = .016, p = .812). There was a positive but non-significant relation between "why to ski" and self-rated competence ($\beta$ = .144, p = .095).

Finally, we looked at whether the number of key learning moments reported was related to either the reported number of challenges within the domain of snow and terrain, self-rated competency or self-rated confidence, controlling for module (baseline and post module 4 were applicable). There was a negligible negative relation between the number of challenges and



the number of key learning moments reported, $\beta$ = .024, p = .874. There was a negligible relation between self-rated competency and number of key learning moments reported, $\beta$ = .069, p = .277. There was a negligible relation between self-rated confidence and number of key learning moments reported, $\tau$ = .071, p = .784. Overall, the number of key learning moments was independent from the kind of challenges reported, confidence gained and self-rated competency.

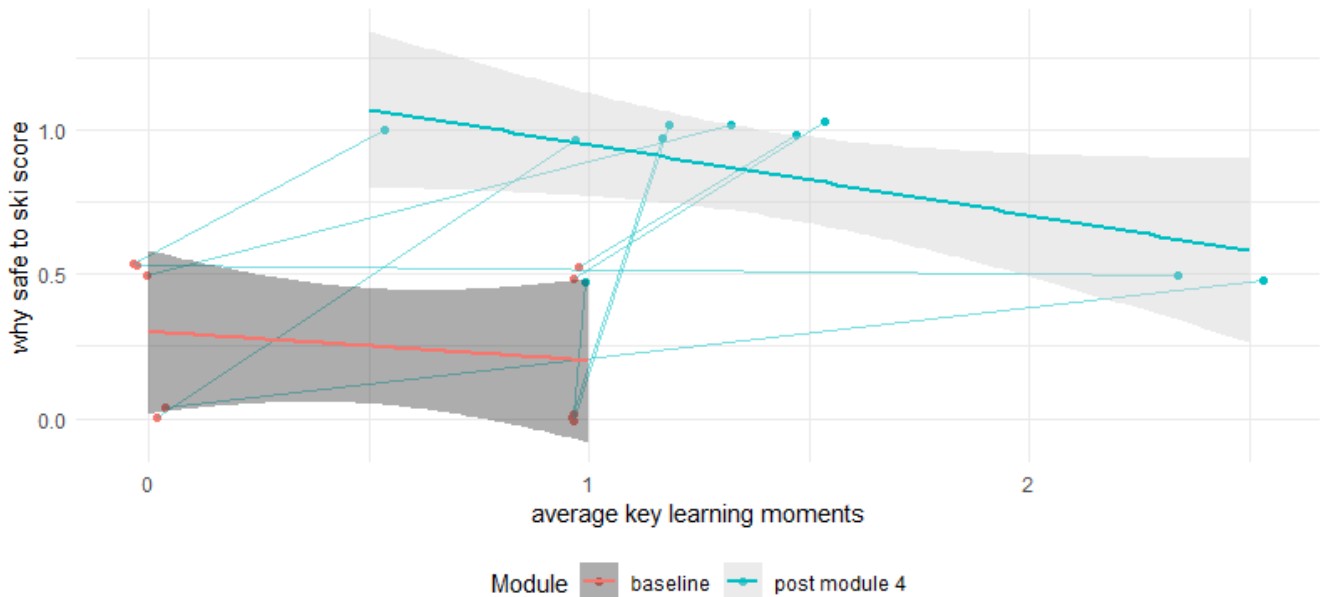

**Figure 3.** Association between key learning moments and reasoning why it is safe to ski. Dots represent individual participants, jittered for better visibility.

In sum, we found some support for hypothesis 1, i.e., during the course knowledge improves but it is not long-lasting for all participants. Hypotheses 2 and 3 were not supported, we did not find that more key learning moments fostered understanding, rather the opposite.

**4 Discussion**

By using multiple scores to assess learning we found that the season-long course fostered understanding of the snow factors, improved the justification of why it is safe to descend, but we also found that learning was not lasting for all participants. At baseline, most participants could not justify why it was safe to descend or correctly distinguish between relevant and irrelevant snow factors. They prefer to rely on the avalanche forecast and did not yet engage in own tracking of the snow and weather conditions. After four modules representing the NF avalanche Level 1 and 2 courses, we found an improvement in knowledge about the snow factors, during planning more high-detail information was gathered, and the explanation why it is safe to



descend was markedly improved. Notably, this did not coincide with a better ability to discriminate relevant from irrelevant factors although there were large individual differences (fig 2 C). One year later, most of the participants fell back to using fewer high detailed planning and also struggled to give a justified explanation grounded in the snow cover analysis of why it was safe to descend. However, knowledge of what the various snow factors represent was still present. This indicates that a
season-long avalanche course is a first step in understanding snow factors, but not sufficient for long-lasting deep learning. Next, we discuss possible explanations, limitations and implications.

**Lack of active experimentation**

Why snow skills did not stick may be explained by comparing the season-long avalanche course to the experiential learning frameworks. Kolb (1984) proposes a cyclical learning process that requires four stages: (1) concrete experience, (2) reflective
observation, (3) abstract conceptualization, (4) and active experimentation. It was attempted to implement all four stages into the course by (1) letting participants concretely experience layering in the snowpack, the anatomy of avalanches, crack initiation and propagation and how snowpack and avalanche characteristics relate to snow factors, by (2) together with participants reflecting on observations post-course day through debriefs, having participants think through and reflect on key learning moments through especially designed reflexivity enhancing surveys, and in-classroom discussions with instructors
and participants, by (3) conceptualizing newly gained knowledge and understanding into approaches toward snowpack assessment, what to look for in the snow, how to identify relevant factors and how to assess and include them in a stability judgements, and by (4) active experimentation by giving the participants concrete tasks to perform on private trips between course modules, e.g. doing snowpack and stability tests, assessing slope stability and avalanche danger level and sharing their tests and judgements on the avalanche bulletin's observation database, discussing and giving feedback both individually and
during in-classroom sessions. The comprehensiveness of the implemented process might explain why we did see a positive relation between reduction of unknown snow factors, increase in correctly identified snow factors, and high "why to ski" score after module 4, but not after the follow up study a year later. Especially stage (4) *active experimentation*, outside the course, may not have happened to the same extent for all participants after the course was finished and when not encouraged by the instructors. An indication that this might have been the case is the goal statement from the AviLog where snow and avalanche
observation was only reported as goal by three participants one year later. Outside the course, many of the participants may have simply not been motivated enough to experiment, i.e., to engage in actively applying and testing their newly acquired knowledge.

Furthermore, if our participants´ reported self-rated confidence is true, then not doing snowpack assessments cannot be explained with a perceived lack of self-efficacy (Bandura, 2011); that low self-efficacy and experienced low ability to do
snowpack assessments makes it more unlikely to engage in this behavior. It might nevertheless be explained by Balent et al. (2016) who find that even though participants of a Level 1 avalanche course reported increased confidence, they also expressed increased uncertainty regarding practical application of theoretical knowledge pointing to a positive relation between increased



knowledge and competence on the one hand, and an increased understanding of the uncertainties and risks related to traveling in potentially hazardous terrain.

Green et al. (2022) discuss that a mismatch between increased confidence and increased skill level to assess snowpack stability can be problematic if it leads to overconfidence (Groves and Varley, 2020), i.e., overestimation of one´s own skills and underestimation of the natural hazard. This is especially true when making decisions in wicked environments where there always is a residual uncertainty regarding the accuracy of one´s own safety judgements. Green et al. (2022) find that those course participants who completed the course but had not yet ventured into the backcountry exhibited greater confidence

compared to those who had already experienced the backcountry and consequently had a better understanding of the challenges involved in practical decision-making. This may also be explained by the description-experience gap (Hertwig and Erev, 2019). In our study, we find a trend towards higher confidence being associated with a higher (better) "why to ski" score, and a positive but non-significant relation between "why to ski" and self-rated competence, which suggests good metacognition (Norman et al., 2019) and alignment between competence and confidence. We also found that the more key learning moments

per module were reported, the less the "why to ski" score improved, i.e., participants who experienced many key learning moments were not yet mastering a justified explanation of why it is safe to descend the critical slope after module 4 (Fig 3). This may indicate two things: (1) participants with lower previous knowledge experience a lot of learning moments, everything is new for them, but a lot of learning does not necessarily correspond to deep understanding and being able to apply the newly acquired knowledge; and (2) it may indicate some awareness of one's own lack of understanding.

If it turns out to be true that not low self-efficacy, but motivation and interest are important for whether backcountry recreationists engage in doing snowpack assessments, this also has important implications for avalanche education. If we want our course participants to learn to make safer decisions based on own snowpack assessments, it is simply not enough to teach these hard skills such as snowpack assessment at avalanche courses. Rather, it is as important to create interest for snow and snowpack assessments and motivate course participants to continue their life-long learning journey by enabling them to learn

outside of and beyond avalanche courses and when they are out touring snowy mountains on their own.

**Quality and authenticity of learning moments**

We only found a negligible relation between the number of key learning moments and the reported number of challenges within the domain of snow and terrain, self-rated competency or self-rated confidence. This suggests that not the quantity of key learning moments but its quality matters for effective learning, and may further be explained by the transformative learning

framework where learning moments need to have certain characteristics, i.e., being disruptive or dislocatory, in order for them to prompt transformative change that leads to people not only changing their understanding and perception of the world, but also being able to act upon this changed understanding (Calleja, 2014). The season-long avalanche course may have simply not provided the course participants with enough high quality dislocatory learning moments that prompted reflection, deep learning and change of practices. This ties in with Biggs (2011) focus on authentic learning. Even though avalanche courses

offer concrete experiences (stage one of Kolb's learning cycle) and attempt to create authentic learning experiences by learning



in the relevant environment, that is, on the mountain in touring situations, it can be argued that it is nevertheless difficult to create these authentic learning situations. There may be several reasons for that.

(1) Since most people will never personally experience being caught by an avalanche and due to the high consequences of getting hurt by avalanches it is ethically not justifiable to expose the learner to real avalanches during an avalanche course. (2)
Choosing good terrain for authentic learning may also be challenging. For example, the complexity of snowpack assessments and spatial variability of the snowpack that create uncertainties regarding stability judgements make it very risky and hard to justify holding avalanche courses in complex avalanche terrain, where wrong judgments may lead to catastrophe. Finding low consequential terrain where participants can make mistakes and learn from them, may also mean that the course terrain is not experienced as authentic or perceived as less relevant for those participants that are eager to venture into (complex) avalanche
terrain. (3) The groups participating in avalanche courses are in most cases artificially created groups, that come together for the course, but do not habitually ski together. Thus, the perceived lack of concrete experience, of relevant learning terrain, and the social artificiality make it difficult to create authentic learning situations during an avalanche course even though most of the courses happen on snowy mountains.

In other words, experiential learning during an avalanche course is not possible in the same way when compared, for instance,
to climbing courses or to Kayaking courses, where falling or the power of the ocean can be experienced concretely, directly and relatively safe compared to being caught by an avalanche. Thus, avalanche course participants may lack the necessary quality of experience especially related to the feedback and consequences of their actions.

One of the intriguing questions arising out of this discussion is how we can replace the experience of real snow avalanches that serve as dislocatory learning moments with what we might call *cognitive and emotional micro avalanches*, that still
stimulate reflection, deep learning, interest and transformative change (see Fjellaksel et al., 2024), but without the potentially disastrous consequences of exposing course participants to the "real thing".

**Time to practice, life-long learning and mentorship**

Landrø et al. (2022) show that both experts and amateur backcountry recreationalists largely consider the same avalanche risk factors to be relevant and recommend an analytical approach toward assessing and judging avalanche risk factors. They also
note that certain skills such as snowpack analysis are hard to learn, but can benefit from an experiential learning approach, citing the direct empirical knowledge, closeness to the action and the events, real and contextual situations (authentic situations) as providing greater learning effect (Beames, 2016). They nevertheless point out that since the learning environment is dynamic (each course and each day is different), it is hard to plan and predict learning because the experienced situations limit what can be learned. This can create challenges, because the time available during recreational avalanche courses will limit the kinds of
situations that can be encountered and learned from. They argue that intended learning outcomes can be difficult to be achieved "without sufficient time" (Landrø et al., 2022, p. 54). This challenge is increased by repetition of the necessary skills taking time and sometimes, being impossible to foster due to the dynamic character of the conditions.





Our study empirically supports this argument by adding that even a lengthy and intensive season-long avalanche course that allows for plenty of repetition and time between modules to digest new knowledge and practice new skills may be too limited
in time to enable all participants to learn about relevant snow factors, how to identify, assess and include these in their own stability judgements, especially if participants are not interested in snow assessments or become interested during the course. Answering the "why to ski" questions, all participants indeed give reasons why skiing a slope was safe. But a year after the course only a few can justify their decisions with information they themselves collected on snow factors and snowpack stability during the trips. Some, but not all participants, are able to understand the snow factors in a way that enables them to relate
them to snowpack stability assessments. This shows that teaching snowpack dynamics is complex and takes a long time. Even interested course participants will need more time than is available during a Level 1, a Level 2 or even a season-long avalanche course. Despite the fact that participants experience a lot of learning, it does not follow automatically that they are also able to apply this newly gained knowledge in their own touring practices. This is food for thought, because with new technology and easy access to avalanche terrain, the competencies needed to make good snow stability judgments may not align with
backcountry recreationalists' exposure to avalanche terrain.

One should nevertheless be cautious to generalize the findings. Our study also shows that each participant's learning trajectory and progress is highly individual, with some benefitting a lot from the course and others to a lesser degree. Still, the fact that we see that most participants struggle to implement the course theory and knowhow of snowpack assessment into their own touring practices a year later suggests that only providing longer avalanche courses may not be enough to teach applied
snowpack assessment. Rather, it seems more prudent to stimulate long term learning by additionally providing shorter refresher and in-depth courses or workshops more often, e.g., on snowpack assessments, especially after a Level 1 and Level 2 course, enabling participants´ continued learning with as little barriers as possible.

Continuous learning and development can also be achieved through mentorship which can play a crucial role in enhancing the effectiveness of experiential learning, particularly in adult education (Koutsoukos, 2020). It encourages personal growth,
supports reflection, and facilitates learning opportunities (Lee, 2007). It is nevertheless challenging to establish mentorships in non-professional settings. Thus, backcountry recreationalists would rely on more informal mentorship relations.

**Method discussion**

**Every day on the mountain is different**

Doing studies like this is methodological challenging beyond ethical and safety considerations. Every module, every tour, the
conditions, how people feel is different each time we are on the mountain. The dynamic socio-ecological context raises the question whether we really can compare the baseline trip to the other modules or the trips a year later. This becomes even more evident when we look at learning being a multi-dimensional process and skills being interrelated and building on each other (Alexander et al., 2009). In our results we see an improvement in the planning abilities in our participants, considering higher detail in the sources they use for planning, e.g. the observation database. We also see indications of good metacognition. This



means that participants might have developed an ability to gain a better understanding of the state of the snowpack already during planning and that these improved planning abilities influence their trip choice, making them choose terrain according to their abilities. Since most of the participants also have stated "ski touring" as goal, not a learning goal during the follow up, participants may have chosen terrain where they thought that doing extensive snow cover analysis was not required under current conditions. In other words, our findings might entirely result from how we measured and rated snowpack assessment

skills.

**Different approaches toward traveling in avalanche terrain and how to assess them**

Løland and Hällgren (2023) show that there is more to sense- and decision making than just checking snow conditions. They argue that there is a multitude of socio-ecological processes happening when for example guides choose where to ski. While guides tend to try and find the best and safest skiing possible (Atkins, 2014) based on current conditions and their clients'

preferences and abilities, avalanche course instructors focus more on where to find the best learning environment for the course participants under current conditions. In this regard, backcountry recreationalists may have a similar mindset to tour guides, trying to find the best skiing that is possible with the group they are a part of under current conditions. In general, they want to avoid avalanche problems, while an instructor's mindset may be comparable to an avalanche observer's mindset trying to find and assess avalanche problems. Instructors focus on finding avalanche problems so participants can learn about snowpack

dynamics in safe terrain. This creates trouble when trying to determine the snowpack assessment skills of course participants when they are allowed to choose their own trips.

As an example, during the follow up study a new form of observation was introduced in the forecast area of Lyngen where one of the post course trips was conducted; Daily online guide meetings in March and April (2024) publishing a thorough expert assessment of the state of the snowpack including recommendations what kind of mind-set backcountry travelers should

employ. Such information is truly valuable when approaching the mountain with the mindset of how to find safe snow and terrain. It saves a lot of time and energy because instead of needing to engage in their own resource demanding analysis of the snowpack, participants could rely on the expert observation and what they perceived as possibly better information (since done by expert guides) compared to what they could have achieved themselves. This line of reasoning also extends to the participant group itself. If some of the group participants are perceived more competent than others, it is rational (Selten and Gigerenzer,

2001) for those participants who think of themselves as less competent to trust the judgements of the more competent group members. Trusting the experts is an approach that on the one hand can save a lot of resources, support fast and frugal decision making (Hertwig and Herzog, 2009) and lead to high quality decisions. On the other hand, it can also lead to a halo error (Kozlowski et al., 1986) when the perceived experts are wrong.

**The factors we do not measure**

Since the course setting is rather artificial with respect to social interactions (not self-chosen tour mates and mostly touring together for the course), with the AviLog as a measurement tool we focus mainly on the domain of snow and terrain,



particularly snow stability and whether participants learned how to identify, assess, and judge relevant snow factors in their own assessments of why a desired slope was safe to ski.

Even though we attempted to measure participants performance regarding snowpack stability assessments, we are not able to
say much about external factors influencing the stickiness of how much participants were able to learn about snow factors´ significance for stability judgments. As an example, the course group consisting of 10 participants was quite large and did not split up during the follow up study. Even though this was possible and mentioned by the researchers. This was surprising since the group at that point had learned (and expressed in participant reports) that such a large group is not beneficial to the sense- and decision-making process, because the fact-finding and discussions take more time. Thus, social aspects as well as group
size might have been contributing to why many participants did not engage in lengthy snow cover analysis and discussion during the follow up trips. They might have simply considered it to not be resource efficient, rather falling back on other strategies. Even though we focus on snow factors and not on human aspects in this paper, this shows that these are important for what and how participants learn. More research into how social cognitive learning (Bandura, 2001) can influence and increase learning effectiveness at avalanche courses may be fruitful.

**Attitudinal fallacy, epistemological power and how to measure applied skills**

One of the big challenges in avalanche education is how to test for the applied skills avalanche course participants acquire. Due to ethical and practical reasons, in many cases, research on these skills is tested on hypothetical scenarios (Green et al., 2022; Landrø et al., 2022; Mannberg et al., 2018; Stephensen et al., 2021). For example, Green et al. (2022) find positive effects of avalanche courses on how course participants perceive risk pre- and post-course by looking at pictures of hypothetical
scenarios. This does not say anything about whether course participants learn to assess the risk and uncertainty related to their own snowpack assessments, and how this translates into what they choose to do in the mountains. There is no guaranteed correspondence between what people say they do, and what they actually do (Argyris and Schön, 1974; Jerolmack and Khan, 2014). The observed effect may be explained by the attitudinal fallacy, the conflation of reported beliefs or attitudes with actual behavior.

Avalanche courses, although well intended, are also epistemological power houses, where particular kinds of knowing and understanding and certain values are taught as being more powerful than others when moving in snow-covered mountains. Epistemological power refers to the influence or control over what is considered knowledge, how it is produced, and who has the authority to define and disseminate it (Segev, 2019). This concept is crucial in understanding the dynamics of knowledge creation and dissemination, as it highlights the power structures that determine what is accepted as truth and whose perspectives
are valued or marginalized. Considering the epistemological power instructors have over course participants, participants´ self-reports and hypothetical tests may not actually test for the applied skills people have learned, which are then argued to influence their risk-perception or even safety behavior but may merely test if avalanche course participants have learned what is accepted and valued to say in certain situations. Our study, even if small in sample size, looks at and quantitatively measures what participants learn and what they do with this knowledge both during the course and when invited back on a self-led ski tour a



575 year after, attempting to avoid the attitudinal fallacy. It makes this study unique, since we do not only assess what people say they do but observe and measure what they actually do both during and a year after the course.

**Participants**

Despite thorough vetting during the recruitment phase, it became apparent early on that there was a discrepancy between the participants' self-reported skills and their actual abilities, as well as their formal avalanche training experiences. This led to a
580 more heterogeneous group than initially expected in terms of skill levels and avalanche education backgrounds. We considered this to be an advantage since it made the group more representative of "regular" avalanche courses where one can expect groups with mixed abilities, previous knowledge and skills.

Due to the two-step recruitment process, where participants had to be part of the CARE panel (Carepanel.no), as well interested and able to participate in a season long avalanche course, we assume that we recruited motivated participants that are not
585 necessarily representative for the larger backcountry population. Anyhow, if snow related skills are not lasting for above average interested course participants, we expect it to be even more difficult for below average interested course participants. Some indication of this can be seen in the goal statement, where only 2 and 3 participants after module 4 and one year later, respectively stated snow and avalanche observation as a goal for the trip.

**Ecological validity**

590 Data on learning was collected during a real-world case study, not a hypothetical scenario, and corroborated by multiple sources of data. Thus, increasing ecological validity of the data, trustworthiness and relevance to the practitioner field.

**5 Implications**

**What does that mean for you as a backcountry recreationalist?**

Focus on continuous learning in authentic situations. Continuously refresh and update. Think of learning in terms of the
595 process, not the result. Continuous learning does not need to be in a formalized way but can also happen through informal learning or mentorship relations. Be aware of the pitfalls of informal learning on the mountain and that not all learning is beneficial. If opting for quality checked avalanche courses, instead of only taking long courses once every 5 or 10 years, plan your avalanche education to be a continuous process. Take shorter courses, but more often, ideally repeating and learning something new each year. Get out there and actively experiment and apply your newly gained skills in terrain where you can
600 make mistakes and learn from them. Make sure your competence matches your confidence and your exposure to avalanche terrain. In addition to having certain trips or mountains as objectives, also set yourself learning goals. Make it your mission to ski those mountains safely and to achieve your learning objectives.



**What does that mean for avalanche course instructors?**

Be conscious that learning on the mountain takes time. It is defined by the socio-ecological context and by the when, the where, the who and the what of experiences and learning situations available to the participants. In addition to the hard and soft skills you teach at your courses, the most important things you can teach participants is to trigger interest and how to keep on learning for themselves after the course has finished. Look for authentic learning situations and encourage participants´ safe experimentation beyond the confines of the courses you teach.

**What does that mean for avalanche course providers?**

People learn a lot during your courses. A lot! Nevertheless, the learning process takes time. Create course offers for backcountry recreationalists beyond standardized Level 1 and Level 2 courses. In addition to offering longer courses, offer shorter courses and workshops. Some can be repeaters, and some can be specialized where people can "dig deeper" into topics such as snowpack dynamics, human dynamics or search and rescue. Train your instructors to facilitate authentic learning situations, make room for and encourage reflection on learning moments and encourage active experimentation under safe conditions.

**Finally, what does that mean for avalanche education research?**

Further research should explore ways to enhance retention and application of avalanche safety knowledge in authentic socio-ecological settings.

**6 Conclusion**

By using multiple assessment scores, we found that the season-long avalanche course improved participants' knowledge about snow factors and their ability to justify safe descents. However, understanding was not seen as reflected in this learning persisting over time. At baseline, participants struggled to distinguish relevant snow factors and track conditions in the field. After course modules following the NF standard for Level 1 and Level 2 avalanche courses, knowledge improved, but participants still found it difficult to identify key factors for safe descent. One year later, detailed knowledge and justified skiing behavior had declined, indicating that while a season-long course is a good introduction, it is insufficient for long-lasting deep learning. In other words, there are no short cuts! An intensive avalanche course will provide a lot of learning and can give you a good head start but to be able to use and apply the knowledge gained in what you do on the mountain, the knowledge and its application needs to be repeated and build upon continuously – only then can it become understanding.



**Data availability:** Materials and data are available at https://osf.io/26x5z/?view_only=02d73beecc9a404bb79f3588b80f5469

**Author contribution**: **TD**: Conceptualization, Methodology, Data Collection, Validation, Investigation, Writing - Original Draft, Writing - Review & Editing, Project administration. **RF**: Conceptualization, Methodology, Data Collection, Validation, Writing - Review & Editing, Supervision. **GP**: Conceptualization, Methodology, Formal analysis, Validation, Investigation, Writing - Original Draft, Writing - Review & Editing, Data Curation, Visualization, Supervision.

**Competing interests: TD** and **RF** own and operate a private company (Tromsø Powder Guides) offering avalanche courses in their spare time.

**Acknowledgements**

Special thanks to all our co-designers and course participants for their contributions and input to the course. We also want to thank Tove Irene Dahl, Tarjei Tveito Skille and Audun Hetland for their thoughtful contributions and support in carrying out

the study. We acknowledge that AI tools such as ChatUiT (an open-source frontend for GPT-3.5/4 by UiT The Arctic University of Norway) and Bing Copilot were used as aid for translation, improving language and literature search.

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
