# Peer review of "Key learning moments as predictors for understanding snowpack dynamics during a season-long avalanche course?"

_EGUsphere, 2024_

## Referee Comment (RC1)

**Review of**
**Key learning moments as predictors for understanding snowpack dynamics during a season-long avalanche course?**
**by T. Dassler et al.**

Frank Techel

Dear authors, dear editor,

please find below my comments on the manuscript. Please note that I lack a background in education and psychology, so my review focuses mainly on the structure and clarity of the *Material and Methods* and *Results* sections. Despite this, I hope my feedback will help enhance the manuscript.

5   This study examines the learning outcomes during and after a season-long avalanche course in Norway. It evaluates the learning outcomes related to snowpack dynamics and stability based on the responses of $\leq 10$ individuals across three course modules, using surveys. Although the topic is highly relevant for enhancing avalanche education, the small sample size may limit the generalizability of the findings. The article is submitted to appear in NHESS as part of the *ISSW 2023 special issue*.

**1   General remarks**

10   – **Size of the data set**: The study relies on a very small data set ($N \leq 10$), which significantly affects the robustness and generalizability of the findings. Given that the study relies on findings from a single avalanche course, the study is more appropriately characterized as a case study, rather than one allowing for broad interpretations. The small sample size is a key limitation that currently is only briefly mentioned (line (L) 573). Please discuss this limitation more thoroughly. To bolster your findings, compare them with results from similar studies. While I am not familiar with this research area and
15   cannot suggest specific literature, a quick search revealed at least one relevant study presented at an ISSW, which could support some of your results (Pfeiffer and Foley, 2006). In my view, comparing your findings with those from similar studies should be the first step in the discussion, before delving into the implications of your results.

– The title of the manuscript suggests that the impact of key learning moments (KLM) on learning outcomes is the primary focus of the study. While the theoretical idea behind KLM is introduced in Sect. 1.3, detailed information about the KLM
20   provided by respondents are not included in the manuscript (L246-249), leaving it unclear what they typically are and how they relate to snowpack stability. Instead, the reader is referred to another paper (currently under review), which I had to read to understand better what KLM are, at least through examples. The analysis exclusively analyzes the number of KLM, but without a clear explanation or examples of what these KLM were and how they relate to learning outcomes or the question, "Why is it safe to ski this slope?", thus limiting the significance of this analysis. - Please reconsider

25     whether KLM should be prominently featured in the title, or, preferably, provide significantly more information to the reader. If KLM were categorized according to a specific annotation scheme, I suggest including that information in the manuscript as well.

- Despite the small data set, the Results section was challenging to follow due to the way data and methods were presented, making it difficult to locate relevant information. Please refer to my specific comments and recommendations below to
30     make these sections more reader-friendly.

- Reading the study made me wonder: why does the study focus exclusively on learning outcomes related to *snowpack dynamics/stability*, and not on important topics related, for instance, to tour planning (beside snow and weather information), terrain choices, group dynamics, and rescue? Are these considered less important? For instance, there is the saying: "If uncertainty or the snowpack is the problem, terrain is the solution." (website) - Please explain to the reader
35     why the focus of this study (or the courses?) was on snowpack dynamics.

- And a rather general comment: The Introduction, and particularly the Discussion, seem rather lengthy. As most of what is introduced and explained is far away from my research expertise, I just wonder whether these could be more focused on the main points relevant to the study.

In the following, please find some more detailed comments. These relate primarily to the *Material and Methods* and the
40 *Results* sections.

**2   Detailed comments**

**2.1   Title**

As mentioned before, if the *key learning moments* aren't explained and analyzed with greater depth, they should probably not be as prominent in the title. Consider mentioning that this is a 'case-study', in the title.

45  ### 2.2   Introduction

In its present structure, the reader has to read to L142-146 and L171-179 to learn about the research objective. Consider moving L142-146 (aims of the study) to the end of the introductory part after L60.

Sections 1.1 to 1.2 provide an extensive literature review on learning. While Sect. 1.3 is clearly relevant for understanding KLM, I feel that parts of Sect.s 1.1 and 1.2 could be more focused on what is relevant for the study.
50 Do you on purpose refer to snowpack dynamics rather than snowpack stability? Or are both relevant? Consider briefly introducing these concepts and why both may be important to teach.

**2.3 Methods**

When reading the Results section, I struggled to easily find the respective information on course modules, when surveys were conducted, which questions were asked, what the possible answer options were, and how answers and answer options were analyzed or grouped. Please make this section much more reader-friendly including the following (but possibly other elements).

(1) A figure, probably a time line, which depicts

- the course modules and the respective labels used in the manuscript,

- when surveys were conducted,

- how many respondents participated in each survey and how many participated in each module,

- the five data sets, which you obtained - and the three, which were analyzed in this study (L233-237),

- . . .

Maybe this doesn't all fit into the same figure.

(2) A table, here or in the Appendix, including

- the survey questions (and maybe a column with an overarching label?) - For instance, in the Results *competence* and *confidence* are analyzed. Firstly, I struggled finding information on how these were addressed. Secondly, in Sect. 2.2.1, I found it difficult to know which question aimed at answering these points.

- the respective answer options (i.e., open text answer) or type of response (i.e., Likert scale)

- (possibly) the respective number of respondents - if that differed between questions,

- possibly the mode of analysis - i.e., text interpretation/annotation by researchers, calculation of median, modeling of scores, . . .

Summarizing this information in figures and/or tables will assist understanding the Results section.

**2.4 Results**

An overall remark, though this is my personal perspective on presenting results accessibly: Consider moving more of the results in tables and figures. In the text, I suggest summarizing the results and highlighting particularly important findings with reference to tables and figures.

Given the small data set and the varying metrics used for different questions (number of participants, proportions correct, proportions not understood, Likert scale, safe-to-ski score), I suggest using simple descriptive measures like median or modal values to describe the findings complementing the modeled values. Presenting these results in a table may make them easier to interpret. Medians could also be shown in the violin plots.

- L288-293: Consider moving this information to a table or a barplot rather than listing proportions for each class individually in the text.

- L294-305: Again, these results could probably be visualized using a barplot making it easier to grasp observed changes from one module to the next.

- L346-365: Consider presenting these results in a table. Maybe association plots would be a way to visualize these relationships too? website

**3 Technical comments**

- To ease readability, consider using hyphens when three nouns are used in a row throughout the manuscript.

- L60: Reading this statement regarding the relevance of what is being taught in avalanche courses from Landrø et al. (2020), I wonder: how did you ascertain that what was being taught was relevant?

- L91: Should the three authors listed be known? Either rephrase or provide references to their works.

- L122 and 136: Why are *Key Memorable Events*, *Dislocatory Moments*, and *Key Learning Moments* written in capital letters?

- L185-188 Provide a reference to these tests, as for instance to the operational guidelines. Alternatively, as they don't seem central to the manuscript, consider shortening the list of tests.

- L267-275: How did you address participants mentioning other reasons for why it was safe to ski? For instance, reasons such as "the slope being small," "skiing one at a time," or "a run-out without any terrain traps" could all be relevant factors that are not related to the snowpack. Wouldn't excluding such reasons potentially obscure the fact that participants made more informed decisions after the season-long course as well?

- L265/269: Out of curiosity: how often did the two raters initially disagree?

- L279: The (c) should probably be a (d)?

- L279: Provide more explanation to the linear mixed models (LMM) and tests. For instance, a brief explanation why LMM is appropriate. Provide a reference to the *R*-package used for LMM calculation.

- L280-283: It is unclear how you analyzed this? Did you also use LMM?

- Section 2.3: Specify the significance level you used when describing results as statistically significant in the Results section.

- L286: I believe that *R Studio* is just the interface software allowing you to interact with *R*. Consider citing the *R* version used for analysis instead.

- L297: What is "safe enough"? Isn't that related to the individual's personal risk perception and acceptance?

- Figure 1: Please describe the sub-figures in the caption according to their appearance in the figure. Please name subfigures (a) to (e). This will also allow to refer the reader to sub-figures in the text.

- Figure 1: It seems unusal to mix a figure and a table in a figure. Check with journal guidelines though I suggest to provide the inserted table as a separate table.

- Figure 2: Why does the violin plot in B show values of 0.25 and 0.75? According to the annotation scheme, only values of 0, 0.5, and 1 are possible.

- L313: Please provide a more detailed explanation of what an effect size of 0.21 signifies. Additionally, it would be helpful to clarify which effect size metric you used. As far as I am aware, journal guidelines typically require writing "0.21" instead of ".21".

- L329-334: The abbreviation $M$ hasn't been introduced before. Initially, I thought it means *median*, until I realised that it probably stands for *module*. - Please introduce all abbreviations clearly when first used.

- L329-334: What are the values after the $M$, for instance on L334 it says $M_{\text{baseline}} = -0.19$. - How can a proportion correct have negative values? If these are values obtained from the model, clearly say so.

- L335-339: You present findings for competence? What about confidence? Were these two correlated?

- L346/354: Both paragraphs start with "We next investigated..." - is the order of investigation relevant? Consider rephrasing.

- L365: Did you analyze the *kind* of challenges? Or their *number*? If the first is the case, how did you do this?

- Figure 3: what are *average* key learning moments? Doesn't this plot show the number of KLM per participant? How can a participant have 1.5 KLM? Or is this due to the jittering in the plot? Please explain that you are showing mean and confidence intervals (derived from LMM, I presume?). - Overall, I really struggle to interpret this plot. Consider to visualize these results using different ways.

- L555-556: The reference to Green et al. (2022) appears to be misspelled. Elsewhere in the text and in the bibliography, it is listed as Greene et al. (2022).

- L593: I suggest removing "for you" as this seems to imply that backcountry recreationalists is a typical audience reading this publication (which I doubt).

- L610: I suggest removing "A lot!"

**135 References**

Landrø, M., Hetland, A., Engeset, R. V., and Pfuhl, G.: Avalanche decision-making frameworks: Factors and methods used by experts, Cold Regions Science and Technology, 170, 102 897, https://doi.org/https://doi.org/10.1016/j.coldregions.2019.102897, 2020.

Pfeiffer, N. and Foley, J.: Skill and knowledge mastery of students in level 1 avalanche courses, in: Proceedings International Snow Science Workshop 2006, Telluride, Colorado, 1–6 Oct 2006, pp. 264–273, https://arc.lib.montana.edu/snow-science/item/935, 2006.

---

## Referee Comment (RC2)

**Review of Key learning moments as predictors for understanding snowpack dynamics during a season-long avalanche course? by T. Dassler et al.**
Anonymous reviewer

***Authors and editors,***

The longitudinal research project described in this manuscript investigated the participant learning on a non-traditionally structured avalanche education course using mixed methods approach. The article focuses on the quantitative analysis of the learning outcome metrics. Data were collected with pre/post surveys, participant activity logs and observer memos, and analyzed to measure the impact of self-reported key learning moments on knowledge about snow stability, and awareness on personal knowledge gaps. The results were non-conclusive in a quantitative sense, however, they provided interesting insight as a case study of avalanche course participants' knowledge and skill retention.

**Overall comment:**

Avalanche education is an understudied field, and the article offers a contribution to our understanding of the efficacy of avalanche courses. The third research hypothesis is clever; approaching education as an opportunity to establish one's own knowledge gaps is brilliant. There is much value in this train of thought, including the authors' recommendations for providing repeated and novel opportunities to life-long learning for mountain travelers(L611).

Despite the potential benefits of this paper, I have some fundamental concerns about the employed methods and the interpretation of the quantitative results. In my opinion, addressing these issues requires a substantial repositioning and rewriting of the manuscript before it can be considered for publishing.

Here are the topical comments on the manuscript focusing mainly on high level points about the methodology and results; I am not providing line-by-line comments for this version.

**Main concern is the limited data set:**

The dataset is very small (8-10 participants) for the application of linear mixed models or Chi-square tests (L279). The number of key learning moments in the dataset is neither included in the article nor the supplemental materials. The authors applied Cramer's V, that is less sensitive to sample size, and pivoted to using effect size when discussing the results. Yet it seems likely that the data set is not viable for Chi-square testing and that extremely small sample can influence the strength of associations used in the results. The results may not be generalizable.

If the article's objective is to share the quantitative results (L19-21; section 2.3), this goal is out of reach due to the limited sample size. The present focus on a quantitative analysis seriously distracts and devalues from more valuable qualitative insights that the study could provide. Hence, reframing the inquiry as the exploration of qualitative insights from the survey data would be more appropriate objective for the manuscript considering the available data set.

**Connecting the dots was difficult:**

The lengthy introduction discusses foundational experiential education literature combined with more current references on effective and transformational learning theories and a short list of relevant articles on avalanche education research. Notably missing is McNeil et al. (2023) that addresses topics of L53-55. Editing the introduction to focus on major theories related to key learning moments, knowledge acquisition, and the role of snowpack information in avalanche decision-making, would prioritize the content that is most relevant for the results and discussion.

It was difficult to follow the connection from hypotheses to conclusions. This may be due to the unclear explanation of the results or the verbose and ambiguous language. If the connections between research questions, theory, data, analysis, and results would tie together more coherently, it would be much easier for readers to understand the conclusions and implications of the study and how well they are grounded in theory and the collected evidence.

Key learning moments
Here are three examples that create convolution related to the title concept, key learning moments (KLM) (Section 1.3), the metric the authors emphasize as a major factor in the efficacy of education:
1. There is no data of KLMs included in Section 2.1 or elsewhere in the paper.
2. In the results section, statements on the variable associations with KLMs are hard to understand (L348-353).
3. X axis on Figure 3 (L370) is labeled "average KLMs", but it is unclear if this number is average KLMs per trip or an overall mean of KLMs per participant during the study period; a clear caption would clarify the figure interpretation.

'Why to ski'
Another mind bender was the 'why to ski' construct the authors use to measure the participants reasoning to decide that it was safe to engage in the activity. The 'why to ski' score is introduced in section2.2 (L267) without a transparent connection to research question about students' ability to apply their knowledge on their own(L154) or their confidence to justify their field decisions (L164). On a different note, the focus on the term 'ski' perpetuates the imbalance in the representation of various mountain activities in avalanche safety research. Even if the dataset consists solely of skiers, the researchers could use more inclusive language when presenting their findings.

Snowpack analysis lessons
An additional content gap is the missing information on what was taught about snowpack analysis. The authors refer to standardized curricula from Norwegian Mountain Forum (L209) but also that the participants had their say about the content (L157).  Including the actual snowpack topics covered, practiced, and reviewed in each module would be informative as an appendix to explain the educational delivery of 23 snowpack factors measured(L164) and analyzed(L260).

**Contributing factors outside the course participation:**

Factors outside the participation in an avalanche education experiment can contribute to the scores in your data; for example, learning moments that happen on personal tour days can build competency. It is possible that the participants engaged in active experimentation outside the course modules (L403). Were these data (L287-293) collected from the participants repeatedly or only once for a baseline? And more generally, did the authors consider how to contextualize the results into the participants' individual lived experiences – not only as participants of four education modules? There is much potential for deep descriptions on the individuals over the repeated interactions with ethnographic approach (Dammler et al., 2023), but that rich viewpoint is missed in this paper. By reframing the paper, the authors could provide a more detailed introduction of their participants, highlighting them as key players in the case study.

**Explanation of how all the parts of the project fit together:**

As a part of a larger research project, the article made references to other papers related to the education experiment (i.e. L135 and 165), which was slightly distracting for this specific piece. I needed to locate and read the other articles mentioned in the text to get the full understanding.  It would have been helpful to have a concise reference to a holistic framework of how the different pieces fit together, perhaps as a flow chart or a table rather than the introductory paragraph in Section 2.2. Sharing the necessary information from the adjacent articles clearly and concisely would improve the thread of the manuscript.

**References:**
McNeil, K., Morgan, J.A., Riggs Meder, L.Y., and Walker, E.R.: Understanding backcountry behaviors after participation in a recreational avalanche course. In: Proceedings of International Snow Science Workshop, Bend, Oregon, 8 October, 2023. pp.1112-1119, https://arc.lib.montana.edu/snow-science/item/3020

---

## Author Comment (AC1)

**Reply to reviewer 2: Anonymous**

Title: Key learning moments as predictors for understanding snowpack dynamics during a season-long avalanche course?
Author(s): Tim Dassler, Richard Fjellaksel, and Gerit Pfuhl
MS No.: egusphere-2024-1533
MS type: Research article
Special issue: Latest developments in snow science and avalanche risk management research – merging theory and practice

Dear Reviewer 2, dear editor,

We are immensely grateful for your time and constructive feedback, which we have used to improve the structure and clarity of the manuscript.

Please find below our reply to your comments.

**Overview of revisions**:

Here we provide a brief overview of the major revisions made to the manuscript based on your comments.

1. We changed the title and included the word 'case study' to address more clearly the characteristics of our study.
2. We reduced the focus on key learning moments (e.g. removed in title, provided simple definition incl. examples).
3. We substantially restructured and rewrote the introduction, methods, findings and discussion section to improve structure and clarity.
4. Suggested literature was included in the discussion section.
5. To make the language more inclusive of the general backcountry population we changed 'safe to ski' assessment to 'safe to descend'. Similar changes were made to the language throughout the paper.
6. We have prepared a revised draft, which you will find in the folder 'Revision_Draft_SafeToDescend' at the following location: https://osf.io/26x5z/?view_only=02d73beecc9a404bb79f3588b80f5469
7. Our line and section numbers refer to the revised version.

**Point-by-point response**:

Below we address each comment raised individually.

Reviewer comments are marked in bold font, our replies in regular font.

**Overall comment:**

**Avalanche education is an understudied field, and the article offers a contribution to our understanding of the efficacy of avalanche courses. The third research hypothesis is clever; approaching education as an opportunity to establish one's own knowledge gaps is brilliant. There is much value in this train of thought, including the authors' recommendations for providing repeated and novel opportunities to life-long learning for mountain travelers (L611).**

**Despite the potential benefits of this paper, I have some fundamental concerns about the employed methods and the interpretation of the quantitative results. In my opinion, addressing these issues requires a substantial repositioning and rewriting of the manuscript before it can be considered for publishing.**

**Here are the topical comments on the manuscript focusing mainly on high level points about the methodology and results; I am not providing line-by-line comments for this version.**

**Main concern is the limited data set:**

**The dataset is very small (8-10 participants) for the application of linear mixed models or Chi-square tests (L279). The number of key learning moments in the dataset is neither included in the article nor the supplemental materials. The authors applied Cramer's V, that is less sensitive to sample size, and pivoted to using effect size when discussing the results. Yet it seems likely that the data set is not viable for Chi-square testing and that extremely small sample can influence the strength of associations used in the results. The results may not be generalizable.**

We thank you for challenging our analysis. Statistical power for repeated measurements is higher for smaller samples than for a similar sized cross-sectional study. As replied to reviewer 1 in four of the outcome variables the effect sizes were large enough that a power analysis yielded over 80% power.
We are aware of the limitations of Cramer's V, however a major advantage of Cramer's V is that it represents the association between two categorical variables as a number between 0 and 1, similar to correlation coefficients and likely easier to understand for most readers (provides familiarity as already Cohen himself noted as a drawback for using Cohen's omega). We cannot use the Phi coefficient as we have larger than 2x2 tables.

**If the article's objective is to share the quantitative results (L19-21; section 2.3), this goal is out of reach due to the limited sample size. The present focus on a quantitative analysis seriously distracts and devalues from more valuable qualitative insights that the study could provide. Hence, reframing the inquiry as the exploration of qualitative insights from the survey data would be more appropriate objective for the manuscript considering the available data set.**

We clarified that this is a case study, e.g. changed the title, and that the focus of this paper is the relation between 'subjective' experience of learning (through key learning moments) and 'objectively' measured learning outcomes (snowpack assessment skills and applied understanding). Since we used an exam-like assessment (AviLog) to measure understanding of snowpack stability, a quantitative approach is justified. We do not claim that our findings are generalizable to the broader winter backcountry population. We discuss that our group of participants also was a highly motivated group.
We further disagree that we cannot report quantitative results given our sample size. Our study is designed as repeated measurement and has more power than a similar sized or larger sized study with two independent groups. A sample size of n=10 has over 80% power to detect large effects (Cohen's D ~1 or larger). See also
https://journalofcognition.org/articles/10.5334/joc.72 and
https://www.ncbi.nlm.nih.gov/pmc/articles/PMC6701714/
It is correct that we did not perform an a priori power analysis, as we had no published effect sizes on which to base the power analysis. However, we did expect large effects from baseline to module 4. We had no prediction for the follow-up one year later. Since we have not pre-registered our hypothesis, we did not include a power analysis.

As mentioned in the manuscript, we do not analyze the qualitative data sets (field observations and focus groups) in this study as this beyond the scope of this paper and will be part of a forthcoming publication.

**Connecting the dots was difficult:**

**The lengthy introduction discusses foundational experiential education literature combined with more current references on effective and transformational learning theories and a short list of relevant articles on avalanche education research.**

**Notably missing is McNeil et al. (2023) that addresses topics of L53-55. Editing the introduction to focus on major theories related to key learning moments, knowledge acquisition, and the role of snowpack information in avalanche decision-making, would prioritize the content that is most relevant for the results and discussion.**

**It was difficult to follow the connection from hypotheses to conclusions. This may be due to the unclear explanation of the results or the verbose and ambiguous language. If the connections between research questions, theory, data, analysis, and results would tie together more coherently, it would be much easier for readers to understand the conclusions and implications of the study and how well they are grounded in theory and the collected evidence.**

We restructured and shortened the introduction for better readability focusing on theories relevant to key learning moments, knowledge acquisition, and the role of snowpack information in avalanche decision-making.

We also clarified the connection between research questions, hypotheses and results and addressed the relation between the experience of learning (key learning moments) and measurement of snowpack assessment skills (AviLog, 'safe to descend' question).

In the introduction we make a point that there are few peer-reviewed studies on avalanche education. But we included the suggested reference (together with other similar studies) in the discussion of our findings.

**Key learning moments**

**Here are three examples that create convolution related to the title concept, key learning moments (KLM) (Section 1.3), the metric the authors emphasize as a major factor in the efficacy of education:**
1. **There is no data of KLMs included in Section 2.1 or elsewhere in the paper.**
2. **In the results section, statements on the variable associations with KLMs are hard to understand (L348-353).**
3. **X axis on Figure 3 (L370) is labeled "average KLMs", but it is unclear if this number is average KLMs per trip or an overall mean of KLMs per participant during the study period; a clear caption would clarify the figure interpretation.**

We agree and thank you for your comments to reduce convolution. We edited the introduction to provide a more coherent definition of key learning moments, including examples, and their importance for experiential learning.

Method section 2.2.1 on key learning moments was revised for clarity.

In the results section findings related were clarified regarding Hypothesis 2 and that the amount of key learning moments increases the request for support, but had no significant association with subjective or objective competencies of assessing snowpack stability (L418-420).

**'Why to ski'**

**Another mind bender was the 'why to ski' construct the authors use to measure the participants reasoning to decide that it was safe to engage in the activity. The 'why to ski' score is introduced in section2.2 (L267) without a transparent connection to research question about students' ability to apply their knowledge on their own(L154) or their confidence to justify their field decisions (L164).**

Important point! Thank you. We clarified the connection between the 'safe to descend' (previously 'why to ski') question and measuring applied understanding of snowpack assessment in the introduction.

We also clarified how we scored participant answers to the 'safe to descend' question based on factors from the systematic snow cover analysis (presence/absence of weak layers, initiation/propagation of cracks, depth of weak layer, thickness and hardness of layer above, etc.)

**On a different note, the focus on the term 'ski' perpetuates the imbalance in the representation of various mountain activities in avalanche safety research. Even if the dataset consists solely of skiers, the researchers could use more inclusive language when presenting their findings.**

This is an important point and something that slipped our attention. Thanks a lot for catching this. We changed 'why to ski' into 'safe to descend' as well as 'ski' and 'skiers' to other terms such as 'backcountry recreationists' 'backcountry travelers' etc. throughout the text.

**Snowpack analysis lessons**

**An additional content gap is the missing information on what was taught about snowpack analysis. The authors refer to standardized curricula from Norwegian Mountain Forum (L209) but also that the participants had their say about the content (L157). Including the actual snowpack topics covered, practiced, and reviewed in each module would be informative as an appendix to explain the educational delivery of 23 snowpack factors measured(L164) and analyzed(L260).**

This is a valid point. We added a new section (2.1.2) where we describe how snowpack analysis was taught during the course. Here we also present the learning goals for snowpack analysis which were the template for scoring the why is it 'safe to descend' answers. We also added an info box (Figure 1) that explains the Systematic Snow-Cover Analysis taught during the course.

**Contributing factors outside the course participation:**

**Factors outside the participation in an avalanche education experiment can contribute to the scores in your data; for example, learning moments that happen on personal tour days can build competency. It is possible that the participants engaged in active experimentation outside the course modules (L403). Were these data (L287-293) collected from the participants repeatedly or only once for a baseline?**

We agree that learning moments and factors outside the course can influence learning and build competency. Participants were also encouraged to explore and experiment outside the course modules as this is a cornerstone of experiential learning. This was done e.g. through 'homework' assignments. See newly added section 2.1.2. However, our main findings indicate that while course participants 'objectively' improve in their snowpack assessment skills during the course (which may be attributed to the course content or other factors), this improvement is not lasting for all a year later. We actively address external factors which we did not measure that might provide alternative explanations of these findings, including the lack of authentic learning experiences during and lack of active experimentation after the course in the discussion section.

**And more generally, did the authors consider how to contextualize the results into the participants' individual lived experiences – not only as participants of four education modules? There is much potential for deep descriptions on the individuals over the repeated interactions with ethnographic approach (Dassler et al., 2023), but that rich viewpoint is missed in this paper. By reframing the paper, the authors could provide a more detailed introduction of their participants, highlighting them as key players in the case study.**

We wholeheartedly agree that the qualitative dimension of learning experiences deserves proper treatment. Since we are here concerned with the analysis of learning outcomes related to snowpack assessment skills regarding their relation to experienced learning moments, we would not be able to do justice to the rich experiences and data collected. This will be the focus of a forthcoming publication that focuses on the qualitative dimensions of the data.

We substantially restructured introduction, methods, findings and discussion to make focus and aim of the paper clearer.

**Explanation of how all the parts of the project fit together:**

**As a part of a larger research project, the article made references to other papers related to the education experiment (i.e. L135 and 165), which was slightly distracting for this specific piece. I needed to locate and read the other articles mentioned in the text to get the full understanding.**

We highly appreciate the extra work invested. By restructuring our introduction, the distractions should now have been vanished.

**It would have been helpful to have a concise reference to a holistic framework of how the different pieces fit together, perhaps as a flow chart or a table rather than the introductory paragraph in Section 2.2. Sharing the necessary information from the adjacent articles clearly and concisely would improve the thread of the manuscript.**

Agreed. We reduced the need to read external articles by providing a rewritten section to the question how we measure learning (subjectively and objectively) in the introduction. This section includes a simpler and more coherent definition of key learning moments and why they are important for learning.

We restructured and rewrote the introduction to improve coherence and clarity. Among other things, we provided a clearer and simpler definition of key learning moments and how we used them in the study.

We included a new figure and a table. Figure 2 shows a timeline of the modules, participants and when the AviLog was filled out (objective measure of snowpack assessment skills). Table 1 is an overview of data collected in each module.

**Conclusion**:

Dear Reviewer 2, we highly appreciate the expertise, time and effort you put into providing constructive feedback. We believe that by addressing your concerns the manuscript has been improved considerably, both in structure, clarity and readability.

We hope that the revisions meet your expectations and concerns.

Kind regards,

Tim Dassler, Richard Fjellaksel and Gerit Pfuhl

---

## Author Comment (AC2)

**Reply to reviewer 1: Frank Techel**

Title: Key learning moments as predictors for understanding snowpack dynamics during a season-long avalanche course?
Author(s): Tim Dassler, Richard Fjellaksel, and Gerit Pfuhl.
MS No.: egusphere-2024-1533
MS type: Research article
Special issue: Latest developments in snow science and avalanche risk management research – merging theory and practice

Dear Frank Techel, dear editor,

We are immensely grateful for your expertise, time and constructive feedback, which we have used to improve the structure and clarity of the manuscript.

Please find below our reply to your comments.

**Overview of revisions**:

Here we provide a brief overview of the major revisions made to the manuscript based on the reviewers' comments.

1. We changed the title and included the word 'case study' to address more clearly the characteristics of our study.
2. We reduced the focus on key learning moments (e.g. removed in title, provided simple definition incl. examples).
3. We restructured and rewrote the introduction, methods, findings and discussion section to improve structure and clarity.
4. Suggested literature was included in the discussion section. Thank you!
5. To make the language more inclusive of the general backcountry population we changed 'safe to ski' assessment to 'safe to descend'.
6. We have prepared a revised draft, which you will find in the folder 'Revision_Draft_SafeToDescend' at the following location: https://osf.io/26x5z/?view_only=02d73beecc9a404bb79f3588b80f5469
7. Line numbers refer to the revised version.

**Point-by-point response**:

Below we address each comment raised individually.

Reviewer comments are marked in bold font, our replies in regular font.

**General remarks**

**Size of the data set: The study relies on a very small data set (N ≤10), which significantly affects the robustness and generalizability of the findings. Given that the study relies on findings from a single avalanche course, the study is more appropriately characterized as a case study, rather than one allowing for broad interpretations.**

-L1-2: We changed the title to address the character of the study being a 'case study'. The new title is: 'Safe to descend? Assessing avalanche course participants' applied understanding of snowpack stability: A case study'

**The small sample size is a key limitation that currently is only briefly mentioned (line (L) 573). Please discuss this limitation more thoroughly.**

We address the limitation of a small study throughout the discussion (e.g. L559f). We would though like to draw attention that our results are not based on a single test but multiple assessments of the same participants, all converging to the same result. Small N but repeated measurements can have more power than cross-sectional studies (large N, single measurement). See https://www.ncbi.nlm.nih.gov/pmc/articles/PMC6701714/

Given our effect size (Cohen's d of 1.92 for safe to descend score baseline vs post module 4) we achieved a power of .9996, or would only have needed n=4 (G power 3.1). Indeed, we had at least a power of .8 for any effect size d > or = 1. This is the case for four of our outcome variables and some of the time periods compared (see Table 2).

**To bolster your findings, compare them with results from similar studies. While I am not familiar with this research area and cannot suggest specific literature, a quick search revealed at least one relevant study presented at an ISSW, which could support some of your results (Pfeiffer and Foley, 2006). In my view, comparing your findings with those from similar studies should be the first step in the discussion, before delving into the implications of your results.**

Great point. Thank you for the suggestion. L424ff: We added a comparison of our findings with two other studies in the discussion and clarified where our study agrees/diverges.

**The title of the manuscript suggests that the impact of key learning moments (KLM) on learning outcomes is the primary focus of the study. While the theoretical idea behind KLM is introduced in Sect. 1.3, detailed information about the KLM provided by respondents are not included in the manuscript (L246-249), leaving it unclear what they typically are and how they relate to snowpack stability. Instead, the reader is referred to another paper (currently under review), which I had to read to understand better what KLM are, at least through examples. The analysis exclusively analyzes the number of KLM, but without a clear explanation or examples of what these KLM were and how they relate to learning outcomes or the question, "Why is it safe to ski this slope?", thus limiting the significance of this analysis.**

**Please reconsider whether KLM should be prominently featured in the title, or, preferably, provide significantly more information to the reader. If KLM were categorized according to a specific annotation scheme, I suggest including that information in the manuscript as well.**

-L1-2: We changed the title, removed the focus of key learning moments, clarified main objective of the study. The new title is: 'Safe to descend? Assessing avalanche course participants' applied understanding of snowpack stability: A case study'

A simplified explanation of key learning moments, including examples is given in the revised introduction. We´ve attempted to clarify their connection to experiential learning and the role KLMs play in our study (subjective experience of learning vs objective measurements of skills). See e.g. L133-153

-Section 2.2.1: We clarified how we analyzed and categorized key learning moments (according to avalanche triangle dimensions).

**Despite the small data set, the Results section was challenging to follow due to the way data and methods were presented, making it difficult to locate relevant information. Please refer to my specific comments and recommendations below to make these sections more reader-friendly.**

We incorporated your suggested changes. Please see detailed replies below.

**Reading the study made me wonder: why does the study focus exclusively on learning outcomes related to *snowpack dynamics/stability*, and not on important topics related, for instance, to tour planning (beside snow and weather information), terrain choices, group dynamics, and rescue? Are these considered less important? For instance, there is the saying: "If uncertainty or the snowpack is the problem, terrain is the**

solution." (website) - Please explain to the reader why the focus of this study (or the courses?) was on snowpack dynamics.

L56ff: An explanation on the importance of snowpack stability and being able to assess it was included in the introduction. Note that becoming better at assessing snowpack stability for being able to know when it was safe enough to enter avalanche terrain was the expressed learning goal of the participants (8 out of 10) (L359-3363). This is measured by the AviLog. We do report tour planning. Since the group setting is artificial (not self-chosen touring partners) we did not report on group dynamics, but discuss our findings in light of the human factor.

**And a rather general comment: The Introduction, and particularly the Discussion, seem rather lengthy. As most of what is introduced and explained is far away from my research expertise, I just wonder whether these could be more focused on the main points relevant to the study.**

We substantially restructured and rewrote the introduction to make clear the focus of the study; the connection between experience of learning/experiential learning and objectively measuring snowpack assessment skills.

Both introduction and discussion got cut on less relevant aspects, but we kept and or expanded on important concepts. For the introduction we have restructured it so that we provide a paragraph on how learning is measured. For the discussion, we stress more the challenges of teaching snowpack stability and stress the strength and limitations of our study.

**Title**

**As mentioned before, if the *key learning moments* aren't explained and analyzed with greater depth, they should probably not be as prominent in the title. Consider mentioning that this is a 'case-study', in the title.**

We changed the title to accommodate the suggestion. Please see above.

**Introduction**

**In its present structure, the reader has to read to L142-146 and L171-179 to learn about the research objective. Consider moving L142-146 (aims of the study) to the end of the introductory part after L60.**

To improve clarity we restructured the introduction. The 'aim of the study' now has its own section. A brief version of why we did the study is stated in L80-81. The aim and hypotheses can be found in L154ff

**Sections 1.1 to 1.2 provide an extensive literature review on learning. While Sect. 1.3 is clearly relevant for understanding KLM, I feel that parts of Sect.s 1.1 and 1.2 could be more focused on what is relevant for the study.**
We restructured these sections and provided a single section (LL82ff) on what learning is, how to measure it, and why subjective experience of learning and its relation to objectively measuring learning outcomes is of interest in this study. We included a more coherent definition of KLMs and some examples.

**Do you on purpose refer to snowpack dynamics rather than snowpack stability? Or are both relevant? Consider briefly introducing these concepts and why both may be important to teach.**

L56-81: We added a section on snowpack stability and why it is important.

We also added an extended section (2.1.2) on how snow cover analysis was taught during the course, including a new Figure 1 summarizing it.

**Methods**

**When reading the Results section, I struggled to easily find the respective information on course modules, when surveys were conducted, which questions were asked, what the possible answer options were, and how answers and answer options were analyzed or grouped. Please make this section much more reader-friendly including the following (but possibly other elements).**

**(1) A figure, probably a time line, which depicts**

 – **the course modules and the respective labels used in the manuscript,**

 – **when surveys were conducted,**

 – **how many respondents participated in each survey and how many participated in each**

 **module,**

 – **the five data sets, which you obtained - and the three, which were analyzed in this study**

 **(L233-237),**

Great suggestion. We included a new figure and a table. Figure 2 shows a timeline of the modules, participants and when the AviLog was filled out (objective measure of snowpack assessment skills). Table 1 is an overview of data collected in each module.

**Maybe this doesn't all fit into the same figure.**

**(2) A table, here or in the Appendix, including**
 – the survey questions (and maybe a column with an overarching label?) - For instance, in the Results *competence* and *confidence* are analyzed. Firstly, I struggled finding information on how these were addressed. Secondly, in Sect. 2.2.1, I found it difficult to know which question aimed at answering these points.

 – **the respective answer options (i.e., open text answer) or type of response (i.e., Likert scale)**

 – **(possibly) the respective number of respondents - if that differed between questions,**

 – **possibly the mode of analysis - i.e., text interpretation/annotation by researchers, calculation of median, modeling of scores, ...**

**Summarizing this information in figures and/or tables will assist understanding the Results section.**

We have provided the requested details in the method section. The survey items have been made available already, please see https://osf.io/26x5z/?view_only=02d73beecc9a404bb79f3588b80f5469

The answer options are stated in the respective sections, e.g. L302ff for self-rated confidence and self-rated comptency (Likert scales), L304ff for the acquired competence, challenges and requested support (text answers). All text answers were scored by the researchers. Calculations are now also reported in more detail, e.g. L292ff

**Results**

**An overall remark, though this is my personal perspective on presenting results accessibly: Consider moving more of the results in tables and figures. In the text, I suggest summarizing the results and highlighting particularly important findings with reference to tables and figures.**

**Given the small data set and the varying metrics used for different questions (number of participants, proportions correct, proportions not understood, Likert scale, safe-to-ski score), I suggest using simple descriptive measures like median or modal values to describe the findings complementing the modeled values. Presenting these results in a table may make them easier to interpret. Medians could also be shown in the violin plots.**

**L288-293: Consider moving this information to a table or a barplot rather than listing proportions for each class individ-**
**L288-293: Consider moving this information to a table or a barplot rather than listing proportions for each class individually in the text.**

We agree with your suggestion. We now provide a new table (Table 2) summarising the descriptives for the three time points and the statistics of the paired tests. We have chosen paired t-tests and adjusted the significance level of the p-value, as we feel for this audience this enhances readability of the statistical results.

**L294-305: Again, these results could probably be visualized using a barplot making it easier to grasp observed changes from one module to the next.**

**L346-365: Consider presenting these results in a table. Maybe association plots would be a way to visualize these relationships too? website**

We have considered tables and figures for illustrating the goals stated by participants, however, with 9 possible goals and multiple answers, 10 participants, and three time points (baseline, module 4, year later), the frequencies are often 0 or 1 for the 9! (36,2880) combinations, that we kept the text of stating the important information. In our case this is that only 2 / 3 indicate snow and avalanche observation as goal of the trip. The same participants did also state e.g. skiing. We have clarified this in the result section. Please see L364ff

**Technical comments**

**To ease readability, consider using hyphens when three nouns are used in a row throughout the manuscript.**

**We considered this, however, after reducing the description of KLMs we found that this is not any longer necessary.**

**L60: Reading this statement regarding the relevance of what is being taught in avalanche courses from Landrø et al. (2020), I wonder: how did you ascertain that what was being taught was relevant?**

We have reformulated it. The course follows the Norwegian NF standard. It is a separate paper or papers to address the question whether what is taught is relevant. Our paper may contribute to this discussion but cannot definitely answer it.

**L91: Should the three authors listed be known? Either rephrase or provide references to their works.**

We removed the reference to these authors as they may not be known in all research fields.

**L122 and 136: Why are *Key Memorable Events*, *Dislocatory Moments*, and *Key Learning Moments* written in capital letters?**

This has been changed in the new version of the manuscript.

**L185-188 Provide a reference to these tests, as for instance to the operational guidelines. Alternatively, as they don't seem central to the manuscript, consider shortening the list of tests.**

We have provided references. See L225-229

**L267-275: How did you address participants mentioning other reasons for why it was safe to ski? For instance, reasons such as "the slope being small," "skiing one at a time," or "a run-out without any terrain traps" could all be relevant factors that are not related to the snowpack. Wouldn't excluding such reasons potentially obscure the fact that participants made more informed decisions after the season-long course as well?**

This was covered in other question items, so participants knew that this item was asking about snow stability. See section 2.2.3

**L265/269: Out of curiosity: how often did the two raters initially disagree?**

12 out of 28 times by 0.5 points. See section 2.2.3 where we added this information, including how raters reached agreement.

**L279: The (c) should probably be a (d)?**

That is correct, thank you!

**L279: Provide more explanation to the linear mixed models (LMM) and tests. For instance, a brief explanation why LMM is appropriate. Provide a reference to the *R*-package used for LMM calculation.**

We have chosen to report t-tests and adjusted the p-value. These tests correspond to follow up tests from a repeated measurement with three time-points. The main results do not change if we use LLMs.

**L280-283: It is unclear how you analyzed this? Did you also use LMM?**

In the new version we have specified that we used t-tests for all comparisons.

**Section 2.3: Specify the significance level you used when describing results as statistically significant in the Results section.**

Thank you, we have included this, see L399

**L286: I believe that *R Studio* is just the interface software allowing you to interact with *R*. Consider citing the *R* version used for analysis instead.**

Analysis was done in R Statistical Software (v4.3; R Core Team 2023). This was added to the manuscript.

**L297: What is "safe enough"? Isn't that related to the individual's personal risk perception and acceptance?**

We did not find the formulation at that line in the original manuscript. We revised the text. In the new manuscript 'safe enough' still appears (e.g. L155), but refers to Landrø´s PhD project and reasoning that one must be able to provide relevant reasons, grounded in snowpack stability assessment.

**Figure 1: Please describe the sub-figures in the caption according to their appearance in the figure. Please name subfigures (a) to (e). This will also allow to refer the reader to sub-figures in the text.**

We refer now to the separate panels in the text.

**Figure 1: It seems unusal to mix a figure and a table in a figure. Check with journal guidelines though I suggest to provide the inserted table as a separate table.**

Thanks for pointing this out. We have not found any detailed information and will change this if requested by the editor / journal.

**Figure 2: Why does the violin plot in B show values of 0.25 and 0.75? According to the annotation scheme, only values of 0, 0.5, and 1 are possible.**

The violin plot is the distribution for all 10 (8) participants, each individual participant has (shown as dot, jittered for visibility) either 0, 0.5 or 1 but the mean (across all participants) can be .25 or .75. All plotted dots are 0, 0.5 or 1. There is no dot at .25 or .75.

**L313: Please provide a more detailed explanation of what an effect size of 0.21 signifies. Additionally, it would be helpful to clarify which effect size metric you used. As far as I am aware, journal guidelines typically require writing "0.21" instead of ".21".**

APA journals require to omit the leading 0. We will change it if the journal requests it.
We used the convention of Cramer's V for df=4 larger than .15 as medium, and larger than .25 as large effect size.

**L329-334: The abbreviation *M* hasn't been introduced before. Initially, I thought it means *median*, until I realised that it probably stands for *module*. - Please introduce all abbreviations clearly when first used.**

We have rewritten this section and this is no longer used, table 2 replaced this information. The M stood for mean.

**L329-334: What are the values after the *M*, for instance on L334 it says $M_{baseline} = -0.19$. - How can a proportion correct have negative values? If these are values obtained from the model, clearly say so.**

We have rewritten this section. The –.19 was a typo. Thanks for catching it.

**L335-339: You present findings for competence? What about confidence? Were these two correlated?**

We now report in Table 2 self-rated competency and self-rated confidence. Yes, self-rated competency and confidence were positively correlated. We report this now.

**L346/354: Both paragraphs start with "We next investigated..." - is the order of investigation relevant? Consider rephrasing.**

This section got rewritten.

**L365: Did you analyze the *kind* of challenges? Or their *number*? If the first is the case, how did you do this?**

Thank you, we now refer to the number of challenges within the snow and terrain domain.

**Figure 3: what are *average* key learning moments? Doesn't this plot show the number of KLM per participant? How can a participant have 1.5 KLM? Or is this due to the jittering in the plot? Please explain that you are showing mean and confidence intervals (derived from LMM, I presume?). - Overall, I really struggle to interpret this plot. Consider to visualize these results using different ways.**

Thank you. As described in the method section, we collated the snow and terrain domain, and collated over 3 modules (six days). To compare it, we used the average (corresponds to the average KLMs per day). As such, a participant can have 1.5 KLMs (per day) if they had for example 6 KLMS for snow and 3 for terrain during module 2, 3 and 4 (1.5 = 9 / 6). this allowed to compare the KLMS at baseline (one day) with the KLMSs experiences post module 4 (six days).

**L555-556: The reference to Green et al. (2022) appears to be misspelled. Elsewhere in the text and in the bibliography, it is listed as Greene et al. (2022).**
Thanks for catching that. Fixed!

**L593: I suggest removing "for you" as this seems to imply that backcountry recreationalists is a typical audience reading this publication (which I doubt).**

Good point. Removed 'for you'.

**L610: I suggest removing "A lot!"**

Agree. Removed.

**Conclusion**:
Dear Frank Techel, we highly appreciate the expertise, time and effort you put into providing constructive feedback. We believe that by addressing your concerns the manuscript has been improved considerably, both in structure, clarity and readability.

We hope that the revisions meet your expectations and concerns.
Kind regards,
Tim Dassler, Richard Fjellaksel and Gerit Pfuhl